# Implicit Optimization Bias of Next-token Prediction in Linear Models

**Christos Thrampoulidis**
Department of Electrical and Computer Engineering
University of British Columbia
Vancouver, Canada
`cthrampo@ece.ubc.ca`

## Abstract

We initiate an investigation into the optimization properties of next-token prediction (NTP), the dominant training paradigm for modern language models. Specifically, we study the structural properties of the solutions selected by gradient-based optimizers among the many possible minimizers of the NTP objective. By framing NTP as cross-entropy minimization across *distinct* contexts, each tied with a *sparse* conditional probability distribution across a finite vocabulary of tokens, we introduce "NTP-separability conditions" that enable reaching the data-entropy lower bound. With this setup, and focusing on linear models with fixed context embeddings, we characterize the optimization bias of gradient descent (GD): Within the data subspace defined by the sparsity patterns of distinct contexts, GD selects parameters that equate the logits' differences of in-support tokens to their log-odds. In the orthogonal subspace, the GD parameters diverge in norm and select the direction that maximizes a margin specific to NTP. These findings extend previous research on implicit bias in one-hot classification to the NTP setting, highlighting key differences and prompting further research into the optimization and generalization properties of NTP, irrespective of the specific architecture used to generate the context embeddings.

## 1   Introduction

Next-token prediction (NTP) has emerged as the go-to paradigm in training modern language models, revolutionizing various applications such as machine translation, text-summarization, and language generation [66]. In NTP, models are trained to predict the most probable token given a sequence of preceding tokens, commonly referred to as the *context*. Concretely, the objective is to learn a mapping from the input context to the probability distribution over the (finite) vocabulary of possible tokens, enabling the model to generate a token that is contextually appropriate [9, 8]. Recently, the NTP paradigm has witnessed remarkable empirical success through its utilization on large-scale deep-learning architectures trained on vast corpora of data [66, 67, 86], leading to unprecedented advances in the field, and the swift integration of these advanced language models into society [62]. Concurrently, researchers have raised critical concerns about robustness, interpretability, and fairness-bias issues arising from our limited understanding of the fundamental operational principles of these models [10, 6]. Despite progress, a comprehensive theory that elucidates the fundamentals of modern language models—including key components like the NTP paradigm and transformer architecture, particularly in terms of optimization and generalization principles—is still lacking.

We initiate an investigation when implicit optimization biases in training language models under the NTP paradigm, particularly in overparameterized regimes where the empirical-loss reaches its lower bound and there is many possible minimizers. To formalize the NTP paradigm, consider

---

38th Conference on Neural Information Processing Systems (NeurIPS 2024).

autoregressive model $q_{\boldsymbol{\theta}}$ parameterized by $\boldsymbol{\theta}$ trained to predict the next-token on sequences of length $T$ using the cross-entropy (CE) loss:

$$\min_{\boldsymbol{\theta}} \; \hat{\mathbb{E}}_{\boldsymbol{z} \sim \mathcal{T}_n} \Big[ \sum_{t \in [T]} -\log \left( q_{\boldsymbol{\theta}}(z_t \,|\, z_1, \ldots, z_{t-1}) \right) \Big]. \tag{1}$$

Here, sequences $\boldsymbol{z} = (z_1, \ldots, z_T)$ consist of tokens $z_t$ from a finite vocabulary $\mathcal{V} = \{1, \ldots, V\}$ and $\hat{\mathbb{E}}$ is expectation over training set $\mathcal{T}_n$ of $n$ such sequences sampled from some underlying true distribution over sequences. Typically, the model $q_{\boldsymbol{\theta}}$ outputs probability of the next token computed via softmax applied on output logits, which are computed by projecting $d$-dimensional embeddings $h_{\boldsymbol{\theta}'}$ to the $V$-dimensional space with a trainable linear decoder $\boldsymbol{W} \in \mathbb{R}^{V \times d}$. Formally, [1]

$$q_{\boldsymbol{\theta}}(z_t \,|\, z_1, \ldots, z_{t-1}) = \mathbb{S}_{z_t}(\boldsymbol{W} h_{\boldsymbol{\theta}'}(z_1, \ldots, z_{t-1})) = \frac{1}{1 + \sum_{\substack{z' \in \mathcal{V} \\ z' \neq z_t}} \exp\left((\boldsymbol{e}_{z'} - \boldsymbol{e}_{z_t})^\top \boldsymbol{W} h_{\boldsymbol{\theta}'}(z_1, \ldots, z_{t-1})\right)}.$$

The CE loss is then minimized over $\boldsymbol{\theta} = (\boldsymbol{W}, \boldsymbol{\theta}')$ using gradient-based methods, e.g. (S)GD, Adam.

We pose the question: *Given training set $\mathcal{T}_n$, what are the structural properties of the weights $\boldsymbol{\theta}$ found by minimizing the NTP objective with gradient-based optimizers?* As in prior research in one-hot supervised classification [2] (e.g. [97, 7, 76, 34]), we specifically target this question in an *overparameterized* setting, where the NTP objective (1) may have an infinite number of solutions, representing an infinite number of models $\boldsymbol{\theta}$ that minimize the training loss. The central challenge is to discern the particular solution the optimizer is inherently biased towards. Since this 'bias' is not explicitly introduced through regularization but is instead ingrained in the training objective and algorithmic structure, it is termed 'implicit bias' [61]. The exploration of implicit bias has a long history in the traditional supervised one-hot classification (see *Related Work in Sec. 6*). In this traditional scenario, the training set comprises feature-label pairs $(\boldsymbol{x}, y)$, where $\boldsymbol{x} \in \mathbb{R}^p$ is a continuous feature, and $y$ represents its unique label. The optimization process minimizes the following training objective (over $\boldsymbol{W}, \boldsymbol{\theta}'$): $\hat{\mathbb{E}}_{(\boldsymbol{x}, y)} \left[ -\log\left(\mathbb{S}_y(\boldsymbol{W} h_{\boldsymbol{\theta}'}(\boldsymbol{x}))\right) \right]$.

At first glance, excluding the sequential format of Eq. (1), the NTP training scenario might seem identical to traditional one-hot prediction: both aim to minimize the same CE loss across models that parameterize probabilities using the softmax of logits. Consider predicting the next token over fixed-length sequences, say sequences of length $t - 1$, via optimizing: $\hat{\mathbb{E}}_{\boldsymbol{z}} \left[ -\log\left(\mathbb{S}_{z_t}(\boldsymbol{W} h_{\boldsymbol{\theta}}(z_1, \ldots, z_{t-1}))\right) \right]$. The context here acts as the feature, and the next token as the label. Recent works [49, 52] draw on such apparent similarities to the traditional one-hot classification paradigm to extrapolate known results from the latter to the NTP setting. However, this comparison overlooks a fundamental, yet critical difference in the nature of the training data that distinguishes these two paradigms (even when the sequential format of Eq. (1) is disregarded): In the traditional setting, each feature (e.g., image) is assigned a single label (e.g., image category). In contrast, in the NTP setting, contexts $z_1, \ldots, z_{t-1}$ of finite length sampled from finite vocabularies are naturally repeated in a (vast) training set, potentially multiple times, each time followed by *different* tokens $z_t$ [73]. Consequently, the NTP paradigm involves training over $m \leq n$ *distinct* (non-repetitive) contexts, each followed by a multitude of possible next tokens, appearing at varying frequencies. For instance, the context `"She is excellent at her role as a"` may be followed by next tokens such as `"doctor,"` `"lawyer,"` `"reviewer,"` or `"mother,"` each with different frequencies. Importantly, certain vocabulary tokens may *not* appear after a given context; e.g., in the above example, tokens like `"run,"` `"and,"` etc., will not follow.

**Model.** We study NTP training over a finite vocabulary employing the following model. Given a large training set of $n$ total sequences, we identify $m \leq n$ *distinct* contexts. Each distinct context $j \in [m]$ is linked to a $V$-dimensional empirical probability vector $\hat{\boldsymbol{p}}_j$, which encodes the frequency with which each vocabulary token follows the context throughout its occurrences in the training set. Crucially, the probability vectors $\hat{\boldsymbol{p}}_j$ are *sparse*, i.e., the support set $\mathcal{S}_j$ of $\hat{\boldsymbol{p}}_j$ satisfies $|\mathcal{S}_j| \ll |\mathcal{V}| = V$. In an extreme where $|\mathcal{S}_j| = 1, \forall j \in [m]$, the probability vector $\hat{\boldsymbol{p}}_j$ becomes one-hot, leading to a scenario reminiscent of the traditional classification setting described earlier. However, such an extreme is essentially improbable in practical language modeling [73]. With this framing, the NTP paradigm is

---

[1]Throughout, $\boldsymbol{e}_v \in \mathbb{R}^V$ is the $v$-th standard basis vector, and $\mathbb{S}_z(\boldsymbol{u}) = \boldsymbol{e}_z^\top \mathbb{S}(\boldsymbol{u})$ the $z$-th entry of softmax output.

[2]In NTP, the ground-truth next token is inherently embedded within the underlying text, thus strictly speaking, it falls under the self-supervised learning paradigm [66]. Yet, the utilization of the CE training objective resembles to supervised training. We leverage this resemblance and regard NTP training as an instance of supervised learning, while also emphasizing how it differs from one-hot encoding supervision.

also related to supervised vision classification with *soft labels*, which advocates for training models on datasets where each example is associated with a vector of soft labels (rather than a one-hot vector), such as by averaging multiple annotators' hard labels [65], knowledge distillation [32] or label smoothing [79]. With this connection, our analysis can also be interpreted (more broadly) as investigating the implicit bias of *sparse* soft-label classification.

## 1.1 Contributions and Organization

**Formulation.** Recognizing the differences between NTP and one-hot classification, we study the question of implicit optimization bias within the NTP setting. To facilitate this, we utilize the model outlined in the previous paragraph and detailed in Sec. 2. For concreteness, our analysis adopts a 'top-down' approach, training only the decoding (also referred to as word-embedding) matrix $W \in \mathbb{R}^{V \times d}$ while keeping context-embeddings fixed. This approach mirrors foundational studies on implicit optimization bias in one-hot classification [76, 34], which first focused on linear models. It allows exploring the complexities of the NTP training objective, distinct from the embedding architecture[3], and while it renders the logits linear and the objective convex, it still poses a technical challenge in terms of determining parameter convergence [76, 34, 37, 60, 38].

**Conditions for reaching entropy.** In Sec. 3, we identify the necessary and sufficient conditions for the logits of the trained model to enable the CE loss to approach its lower bound, the empirical conditional entropy. We introduce two conditions: $\text{NTP}_{\mathcal{H}}$-compatibility and NTP-separability, which impose constraints on mutually orthogonal subspaces that are determined by the *sparsity patterns* of *distinct* contexts within the dataset. These conditions determine the necessary and sufficient overparameterization a model needs to achieve the empirical entropy lower bound during training.

**Margin in NTP setting.** Motivated by the NTP-separability condition, we introduce a margin concept for NTP in Sec. 4, which extends the classical definition of margin used in one-hot supervised classification [88]. We further establish the relevance of this new margin notion for optimization by demonstrating that a decoder maximizing the NTP-margin, denoted as $W^{\text{mm}}$, guides the directional convergence of the ridge-regularized CE minimizer, $\widehat{W}_\lambda$, as the regularization parameter $\lambda \to 0$.

**Implicit bias of GD.** We establish that $W^{\text{mm}}$ also determines the implicit bias of gradient descent (GD) iterates in Sec. 5. Specifically, in the limit of iterations $k \to \infty$, the GD iterates grow undoubtedly in norm and converge to a finite $W^\star$ within a data subspace $\mathcal{F}$, while simultaneously aligning with $W^{\text{mm}}$ in the complementary subspace $\mathcal{F}^\perp$. The finite component $W^\star \in \mathcal{F}$ solves a system of linear equations associated with the $\text{NTP}_{\mathcal{H}}$-compatibility condition.

Finally, we numerically verify these findings and discuss related and future work in Secs. 6 and 7. Additional experiments, further related work and detailed proofs are in the appendix.

## 2 Setup

Let vocabulary $\mathcal{V} = [V] := \{1, \ldots, V\}$ represent a set of $V$ tokens (e.g. words) and $\boldsymbol{z}_{1:t} = (z_1, \ldots, z_t)$ denote sequence of $t$ tokens $z_t \in \mathcal{V}$. To simplify presentation, we focus on predicting the $T$-th token $z_T$ given contexts $\boldsymbol{z}_{<T} := \boldsymbol{z}_{1:T-1}$ of fixed length, and we further let $\boldsymbol{x} = \boldsymbol{z}_{<t}$ denote the context and $z$ denote the last token. See App. C for straightforward extension to the sequential format of Eq. (1).

We assume access to a training set consisting of $n$ sequences $\mathcal{T}_n := \{(\boldsymbol{x}_i, z_i)\}_{i \in [n]}$, with $\boldsymbol{x}_i \in \mathcal{X} := \mathcal{V}^{T-1}$ and $z_i \in \mathcal{V}$. Let $h : \mathcal{X} \to \mathbb{R}^d$ an embedding map that maps contexts (i.e., sequences of $T-1$ tokens) to $d$-dimensional embeddings. The map $h$ can be parameterized (e.g. by a transformer [90] or an LSTM [5]), but this paper assumes that it is fixed. The next-token is predicted via a linear model $f_{\boldsymbol{W}} : \mathcal{X} \to \mathbb{R}^V$ parameterized by decoding matrix $\boldsymbol{W} \in \mathbb{R}^{V \times d}$, such that $f_{\boldsymbol{W}}(\boldsymbol{x}) = \boldsymbol{W} h(\boldsymbol{x})$. When the model output passes through a softmax, it defines the model's probability mass function for the next-token prediction, given as $\hat{q}_{\boldsymbol{W}}(\cdot | \boldsymbol{x}) = \mathbb{S}(f_{\boldsymbol{W}}(\boldsymbol{x}))$, where $\mathbb{S}(\cdot) : \mathbb{R}^V \to \Delta^{V-1}$ is the softmax and $\Delta^{V-1}$ is the $V$-dimensional simplex. The decoder is trained by minimizing the empirical CE loss $\text{CE}(\boldsymbol{W}) := \frac{1}{n} \sum_{i \in [n]} -\log(\hat{q}_{\boldsymbol{W}}(z_i | \boldsymbol{x}_i))$.

**Distinct sequences and next-token distributions.** Given dataset $\mathcal{T}_n$ we denote $\bar{\boldsymbol{x}}_1, \ldots, \bar{\boldsymbol{x}}_m$ the $m \leq n$ *distinct* contexts among the (large number of) total $n$ contexts $\boldsymbol{x}_1, \ldots, \boldsymbol{x}_n$ within $\mathcal{T}_n$. Let $\hat{\pi}_j$

---

[3]NTP is widely used across various modern language modeling architectures, including transformers [66, 67], state-space models [26, 27], and LSTMs [5].

be the empirical probability of distinct context $\bar{\boldsymbol{x}}_j$. That is, $1 \le n \cdot \hat{\pi}_j \le n$ is the number of contexts $\boldsymbol{x}_i$ that equal $\bar{\boldsymbol{x}}_j$. Furthermore, for each distinct context $\bar{\boldsymbol{x}}_j$, $j \in [m]$ let $\hat{\boldsymbol{p}}_j \in \Delta^{V-1}$ denote the probability vector of conditional next-token distribution, i.e., $\hat{p}_{j,z} := \hat{p}(z|\bar{\boldsymbol{x}}_j)$, $z \in \mathcal{V}, j \in [m]$. In other words, $n \cdot \hat{\pi}_j \cdot \hat{p}_{j,z}$ is the number of occurences of token $z$ as a follow-up to context $\bar{\boldsymbol{x}}_j$. Finally, we denote the support set and size of the *support set* of these conditional distributions as $\mathcal{S}_j := \{z \in \mathcal{V} \,|\, \hat{p}_{j,z} > 0\}$ and $S_j := |\mathcal{S}_j|$. Tokens $z \in \mathcal{S}_j$ and $v \notin \mathcal{S}_j$ are referred to as 'in-support' and 'out-of-support' respectively. Onwards, we implicitly assume that "*not all tokens are likely after every context*," i.e. $\exists j \in [m]$ such that $S_j < V$. This mild assumption is naturally satisfied in language modeling under rich enough vocabulary. With this notation, [4] we can express the NTP training loss as

$$\mathrm{CE}(\boldsymbol{W}) = - \sum_{j \in [m]} \hat{\pi}_j \sum_{z \in \mathcal{V}} \hat{p}_{j,z} \log \left( \mathbb{S}_z(\boldsymbol{W} h(\bar{\boldsymbol{x}}_j)) \right) = - \sum_{j \in [m]} \hat{\pi}_j \sum_{z \in \mathcal{S}_j} \hat{p}_{j,z} \log \left( \mathbb{S}_z(\boldsymbol{W} \bar{\boldsymbol{h}}_j) \right), \quad (2)$$

where, in the last line we defined the shorthand $\bar{\boldsymbol{h}}_j = h(\bar{\boldsymbol{x}}_j)$. Similarly, we let $\boldsymbol{h}_i = h(\boldsymbol{x}_i), i \in [n]$. With some abuse of notation, we then obtain the following equivalent descriptions of the training set

$$\{(\boldsymbol{x}_i, z_i)\}_{i \in [n]} =: \mathcal{T}_n \equiv \mathcal{T}_m := \{(\bar{\boldsymbol{h}}_j, \hat{\pi}_j, \hat{p}_{j,z \in \mathcal{V}})\}_{j \in [m]}$$

that emphasizes *distinct* contexts and their respsective sparse next-token probability distributions.

**Entropy.** The *empirical $T$-gram entropy* (referred to hereafter as entropy for simplicity) of the dataset is [74, 73]: $\mathcal{H}_T := \mathcal{H} := \hat{\mathbb{E}}_{(\boldsymbol{x},z) \sim \mathcal{T}_n} \left[ -\log(\hat{p}(z|\boldsymbol{x})) \right] = -\sum_{j \in [m]} \sum_{z \in \mathcal{S}_j} \hat{\pi}_j \hat{p}_{j,z} \log(\hat{p}_{j,z})$ . It lower bounds the CE loss since $\mathrm{CE}(\boldsymbol{W}) = \mathcal{H} + \mathrm{KL}(\hat{\boldsymbol{p}} \,\|\, \hat{\boldsymbol{q}}_{\boldsymbol{W}})$ and the KL divergence is nonnegative.

## 3 When can the NTP-loss reach the entropy lower-bound?

The first question we ask is: Under what conditions on the training data can the CE loss reach its entropy lower-bound? By the entropy lower-bound, $\mathrm{CE}(\boldsymbol{W}) = \mathcal{H} \Leftrightarrow \mathrm{KL}(\hat{\boldsymbol{p}} \,\|\, \hat{\boldsymbol{q}}_{\boldsymbol{W}}) = 0$ iff for all $j \in [m]$ and all $z \in \mathcal{V}$: $\hat{q}_{\boldsymbol{W}}(z|\bar{\boldsymbol{x}}_j) = \hat{p}_{j,z}$. Equivalently, for all $j \in [m]$:

$$\mathbb{S}_z(\boldsymbol{W} \bar{\boldsymbol{h}}_j) = \hat{p}_{j,z}, \quad \forall z \in \mathcal{S}_j, \quad (3a)$$

$$\mathbb{S}_v(\boldsymbol{W} \bar{\boldsymbol{h}}_j) = 0, \quad \forall v \notin \mathcal{S}_j. \quad (3b)$$

Beginning with (3a), this requires[5] the training data to satisfy the NTP$_{\mathcal{H}}$-compatibility condition defined below.

**Definition 1** (NTP$_{\mathcal{H}}$-compatible). *Let $\boldsymbol{e}_v$ denote the $v$-th standard basis vector in $\mathbb{R}^V$. We say that training data $\mathcal{T}_m$ are NTP-entropy-compatible if there exists $V \times d$ matrix $\boldsymbol{W}^{\mathrm{p}}$ satisfying:*

$$\forall j \in [m], z \ne z' \in \mathcal{S}_j \,:\, (\boldsymbol{e}_z - \boldsymbol{e}_{z'})^\top \boldsymbol{W}^{\mathrm{p}} \bar{\boldsymbol{h}}_j = \log(\hat{p}_{j,z}/\hat{p}_{j,z'}) . \quad (4)$$

We comment on the independence of the constraints: Fix any $j \in [m]$. Then, the set of constraints (as expressed in Eq. (4)) for all $z \ne z' \in \mathcal{S}_j$ (yielding $\binom{S_j}{2}$ constraints in total) is equivalent to the set of the same constraints for any anchor $z_j \in \mathcal{S}_j$ and $z' \ne z_j \in \mathcal{S}_j$, i.e., an effective total of $S_j - 1$ linearly independent constraints for each $j \in [m]$. Additionally, note that the system of equations in Eq. (4) constrains $\boldsymbol{W}^{\mathrm{p}}$ with respect to a specific subspace of $V \times d$ matrices:

$$\mathcal{F} = \mathrm{span}\left( \left\{ (\boldsymbol{e}_z - \boldsymbol{e}_{z'}) \bar{\boldsymbol{h}}_j^\top \,:\, z \ne z' \in \mathcal{S}_j, j \in [m] \right\} \right), \quad (5)$$

that is defined in terms of context embeddings and their respective support sets. Assuming Eqs. (4) have a solution, we denote the *unique* solution *within the subspace* $\mathcal{F}$ as $\boldsymbol{W}^\star \in \mathcal{F}$ for later reference [6].

Next, we examine Eq. (3b), which requires softmax outputs be zero for tokens that never occur following a fixed context throughout the dataset. Due to the strict positivity of softmax, the constraint is never satisfied for *finite $\boldsymbol{W}$*. Thus, for all finite $\boldsymbol{W}$, there exists a gap between the cross-entropy loss and its lower bound, i.e., $\mathrm{CE}(\boldsymbol{W}) > \mathcal{H}$. Yet, it is possible to approach entropy as the norm of the weights $\boldsymbol{W}$ grows, provided that weights move in the appropriate direction formalized below.

---

[4] A complete list of notations is also given in Appendix D.

[5] It will be see below, and can be easily checked by the reader, this condition alone is insufficient; the NTP-separability condition in Defn. 2 is also needed.

[6] If Eqs. (4) have a solution, say $\boldsymbol{W}_1$, every other solution takes the form $\boldsymbol{W}^p = \boldsymbol{W}_1 + \boldsymbol{W}_{\mathrm{null}}$, where $\boldsymbol{W}_{\mathrm{null}}$ is orthogonal to $(\boldsymbol{e}_z - \boldsymbol{e}_{z'})\bar{\boldsymbol{h}}_j^T : z \ne z' \in \mathcal{S}_j, j \in [m]$. Thus, $\boldsymbol{W}_{\mathrm{null}} \in \mathcal{F}^\perp$ is in the orthogonal complement of $\mathcal{F}$.

**Definition 2** (NTP-separable). *We say that training data $\mathcal{T}_m$ are NTP-separable if there exists $V \times d$ matrix $\boldsymbol{W}^{\mathrm{d}}$ satisfying the following:*

$$\forall j \in [m], z \neq z' \in \mathcal{S}_j \ : \ (\boldsymbol{e}_z - \boldsymbol{e}_{z'})^\top \boldsymbol{W}^{\mathrm{d}} \bar{\boldsymbol{h}}_j = 0 \tag{6a}$$

$$\forall j \in [m], z \in \mathcal{S}_j, v \notin \mathcal{S}_j \ : \ (\boldsymbol{e}_z - \boldsymbol{e}_v)^\top \boldsymbol{W}^{\mathrm{d}} \bar{\boldsymbol{h}}_j \geq 1 \,. \tag{6b}$$

As before, it is easy to see that the constraints in (6) can be equivalently expressed by enforcing (6a) and (6b) for an anchor $z_j \in \mathcal{S}_j$ and all $z' \in \mathcal{S}_j \setminus \{z_j\}$ and $v \notin \mathcal{S}_j$, respectively. Consequently, there exist effectively $V - 1$ linearly independent constraints per context $j \in [m]$.

We now discuss the interpretation of these constraints. The subspace constraints in Eq. (6a) project $\boldsymbol{W}^{\mathrm{d}}$ onto the subspace $\mathcal{F}^\perp$, which is the orthogonal complement of the subspace $\mathcal{F}$ defined in (5). This leaves the softmax probabilities of possible next tokens (in set $\mathcal{S}_j$) intact, and fully determined by $\boldsymbol{W}^{\mathrm{p}}$ as per the NTP$_{\mathcal{H}}$-compatibility condition. Formally, $\boldsymbol{W}^{\mathrm{p}} + \boldsymbol{W}^{\mathrm{d}}$ continues satisfying (4). Moving on the halfspace constraints in (6b), we can interpret these using Kesler's construction as enforcing linear separability in the space $\mathbb{R}^{V \times d}$ [30]: Each $d$-dimensional context embedding $\bar{\boldsymbol{h}}_j$ is mapped to $S_j(V - S_j)$ higher-dimensional points $(\boldsymbol{e}_z - \boldsymbol{e}_v)\bar{\boldsymbol{h}}_j^\top, z \in \mathcal{S}_j, v \notin \mathcal{S}_j$. These points collectively for all $j \in [m]$ must lie within the interior of the same halfspace induced by the hyperplane $\langle \boldsymbol{W}^{\mathrm{d}}, \cdot \rangle = 0$. Refer to Fig. 1(Left) and its caption for an alternative interpretation of the rows of $\boldsymbol{W}^{\mathrm{mm}}$ as word-embeddings in $\mathbb{R}^d$ (illustration in $d = 2$).

The impact of NTP-separability on the softmax probabilities can be understood algebraically by considering $\boldsymbol{W}_\gamma := \gamma \boldsymbol{W}^{\mathrm{d}}$ and $v \notin \mathcal{S}_j$. We have:

$$\mathbb{S}_v(\boldsymbol{W}^\gamma \bar{\boldsymbol{h}}_j) = \Big( \sum_{z \in \mathcal{S}_j} e^{\gamma(\boldsymbol{e}_z - \boldsymbol{e}_v)^\top \boldsymbol{W}^{\mathrm{d}} \bar{\boldsymbol{h}}_j} + \sum_{v' \notin \mathcal{S}_j} e^{\gamma(\boldsymbol{e}_{v'} - \boldsymbol{e}_v)^\top \boldsymbol{W}^{\mathrm{d}} \bar{\boldsymbol{h}}_j} \Big)^{-1}$$

$$\leq \Big( \sum_{z \in \mathcal{S}_j} e^{\gamma(\boldsymbol{e}_z - \boldsymbol{e}_v)^\top \boldsymbol{W}^{\mathrm{d}} \bar{\boldsymbol{h}}_j} \Big)^{-1}$$

$$\leq e^{-\gamma}, \tag{7}$$

where the first inequality removes non-negative exponential terms and the second one follows from (6b). The upper bound above approaches 0 as $\gamma \to \infty$, thus (3b) holds asymptotically in $\gamma$.

Taking into account the observations made above, the satisfaction of both conditions guarantees convergence of the cross-entropy loss CE to $\mathcal{H}$. This is formalized in the proposition below.

**Proposition 1.** *Assume training data $\mathcal{T}_m$ is NTP$_{\mathcal{H}}$-compatible and NTP-separable, with the respective matrices $\boldsymbol{W}^{\mathrm{p}}$ and $\boldsymbol{W}^{\mathrm{d}}$ satisfying conditions (4) and (6). While all finite $\boldsymbol{W}$ satisfy $\mathrm{CE}(\boldsymbol{W}) > \mathcal{H}$, it holds for $\boldsymbol{W}^\gamma = \boldsymbol{W}^{\mathrm{p}} + \gamma \cdot \boldsymbol{W}^{\mathrm{d}}$ that $\mathrm{CE}(\boldsymbol{W}^\gamma) \xrightarrow{\gamma \to +\infty} \mathcal{H}$.*

Hence, CE approaches its lower-bound in the limit of a *direction* $\overline{\boldsymbol{W}^{\mathrm{d}}} := \boldsymbol{W}^{\mathrm{d}} / \|\boldsymbol{W}^{\mathrm{d}}\|$ and *offset* $\boldsymbol{W}^{\mathrm{p}}$ satisfying the constraints of NTP-separability and NTP-compatibility, respectively. In other words, parameter weights $\boldsymbol{W}$ that minimize the CE loss consist of two components: a finite projection $\boldsymbol{W}_{\mathcal{F}} := \mathcal{P}_{\mathcal{F}}(\boldsymbol{W}) = \boldsymbol{W}^\star$ onto the data subspace $\mathcal{F}$ and an infinite-norm component onto the orthogonal complement $\mathcal{F}^\perp$ in the direction of $\boldsymbol{W}^{\mathrm{d}}$.

Finally, we note that while Defns. 1 and 2 are stated for linear models, they naturally extend to a more general formulation for *nonlinear* models. Specifically, consider NTP-separability (similar for NTP-compatibility): the general conditions require that both the decoder weights $\boldsymbol{W}$ and model weights $\boldsymbol{\theta}$, which parameterize the embeddings $\bar{\boldsymbol{h}}_j = h_{\boldsymbol{\theta}}(\bar{\boldsymbol{x}}_j)$, must satisfy Eq. (6) simultaneously.

### 3.1 The role of overparameterization

We show that overparameterization provides a sufficient condition for the solvability of Eqs. (4) and (6). Start with the halfspace constraints in Eq. (4) for NTP$_{\mathcal{H}}$-compatibility. These can be compactly expressed as $\boldsymbol{E}_{j,z_j} \boldsymbol{W}^{\mathrm{p}} \bar{\boldsymbol{h}}_j = \boldsymbol{a}_{j,z}$, where $\boldsymbol{E}_{j,z_j} \in \mathbb{R}^{(S_j-1) \times V}$ has rows $\boldsymbol{e}_{z_j} - \boldsymbol{e}_{z'}$ and $\boldsymbol{a}_{j,z_j} \in \mathbb{R}^{(S_j-1)}$ has entries $\log(\hat{p}_{j,z_j}/\hat{p}_{j,z'})$ for some anchor $z_j \in \mathcal{S}_j$. Now, since the rows of $\boldsymbol{E}_{j,z_j}$ are linearly independent, the question becomes equivalently that of determining when $\boldsymbol{W}^{\mathrm{p}}[\bar{\boldsymbol{h}}_1, \ldots, \bar{\boldsymbol{h}}_m] = [\boldsymbol{E}_{1,z_1}^\dagger \boldsymbol{a}_{1,z_1}, \ldots, \boldsymbol{E}_{m,z_m}^\dagger \boldsymbol{a}_{m,z_m}]$ has a solution. This is always the case when $d > m$ and the $d \times m$

embedding matrix $\bar{\boldsymbol{H}} = [\bar{\boldsymbol{h}}_1, \ldots, \bar{\boldsymbol{h}}_m]$ is full rank $(m)$. Then, there exists $\boldsymbol{W}^{\mathrm{p}}$ such that condition (4) holds. In fact, $\bar{\boldsymbol{H}}^\top$ has a nullspace, implying the existence of an infinite number of solutions to (4). These solutions take the form $\boldsymbol{W}^{\mathrm{p}} = \boldsymbol{W}^\star + \boldsymbol{W}_\perp^{\mathrm{p}}$, where $\boldsymbol{W}^\star \in \mathcal{F}$ is the unique solution onto the subspace, and $\boldsymbol{W}_\perp^{\mathrm{p}} \in \mathcal{F}^\perp$.

In contrast to (4), the constraints in (6) involve linear inequalities. However, a sufficient proxy for feasibility in this case is that the corresponding system of equations (instead of inequalities) has a solution. By following the exact same argument as before, we arrive at the same sufficient conditions for the existence of a solution $\boldsymbol{W}^{\mathrm{d}}$. We summarize these findings.

**Lemma 1** (Overparameterization implies NTP-separability). *Assume overparameterization $d > m$ and full-rank embedding matrix $\bar{\boldsymbol{H}} \in \mathbb{R}^{d \times m}$. Then, there exists an infinite number of solutions $\boldsymbol{W}^{\mathrm{p}}$ and $\boldsymbol{W}^{\mathrm{d}}$ that satisfy conditions (4) and (6), respectively.*

Thus, $d > m$, [7] which also generically favors full-rankness of the embedding matrix [92], implies both NTP$_{\mathcal{H}}$-compatibility and NTP-separability. Combined with Prop. 1, it also implies that there are infinitely many possible directions $\boldsymbol{W}^{\mathrm{d}}$ along which the NTP loss approaches $\mathcal{H}$, motivating the implicit-bias question: For a specific iterative algorithm aimed at minimizing the NTP loss, which direction does it prefer? We will address this question in the remainder of the paper.

**Remark 1.** *In the trivial case where $S_j = 1, \forall j \in [m]$ (one-hot classification), the entropy lower bound is zero and is attained iff the data is linearly separable. Indeed, $\mathcal{F}$ reduces to the empty set, and NTP-separability simplifies to traditional multiclass separability. For binary classification, [20] showed that $d/m > 1/2$ is sufficient and necessary for data in general position to be linearly separable. More recently, several works have extended this analysis to structured (random) data, including [12, 71, 57, 54]. The exact threshold in corresponding mutliclass settings is more intricate, but [19, 81, 11] have made progress in this direction. An interesting question is determining exact thresholds for NTP-separability, which would improve upon the sufficient condition of Lemma 1.*

## 4    Regularization path

This section investigates the implicit bias of NTP by examining the minimization of CE loss through iterates defined as follows for an increasing sequence of positive regularization parameters $B$:

$$\widehat{\boldsymbol{W}}_B := \arg\min_{\|\boldsymbol{W}\| \le B} \mathrm{CE}(\boldsymbol{W}). \tag{8}$$

This involves minimizing a strictly convex function in a bounded domain; thus, $\widehat{\boldsymbol{W}}_B$ is unique. This section's main result characterizes the limit of $\widehat{\boldsymbol{W}}_B$ as $B \to \infty$ under NTP-separability/compatibility. Before that, we first define the next-token prediction support-vector machines (SVM) problem.

**Definition 3** (NTP-SVM). *Given NTP-separable training set $\mathcal{T}_m$, NTP-SVM solves the following:*

$$\boldsymbol{W}^{\mathrm{mm}} := \arg\min_{\boldsymbol{W}} \ \|\boldsymbol{W}\| \qquad \textit{subj. to } \boldsymbol{W} \in \mathbb{R}^{V \times d} \textit{ satisfying (6a) and (6b)}. \qquad \text{(NTP-SVM)}$$

This is a strongly convex quadratic program with $mV - \sum_{j \in [m]} S_j$ linear inequality and $\sum_{j \in [m]} S_j - m$ linear equality constraints. Its solution can be also defined as the classifier that maximizes margin between in and out-of -support tokens while being constrained on the orthogonal compelemnt $\mathcal{F}^\perp$:

$$\overline{\boldsymbol{W}^{\mathrm{mm}}} = \arg\max_{\|\boldsymbol{W}\|=1, \boldsymbol{W} \in \mathcal{F}^\perp} \min_{j \in [m], z \in \mathcal{S}_j, v \notin \mathcal{S}_j} (\boldsymbol{e}_z - \boldsymbol{e}_v)^\top \boldsymbol{W} \bar{\boldsymbol{h}}_j.$$

It turns out this direction determines the preferred limiting direction of the regularization path.

**Theorem 1** (Implicit bias of the regularization-path). *Assume training data $\mathcal{T}_m$ is NTP$_{\mathcal{H}}$-compatible and NTP-separable. Let $\widehat{\boldsymbol{W}}_B$ be defined as in (8). Then, it holds that $\lim_{B \to \infty} \left\langle \frac{\widehat{\boldsymbol{W}}_B}{\|\widehat{\boldsymbol{W}}_B\|}, \frac{\boldsymbol{W}^{\mathrm{mm}}}{\|\boldsymbol{W}^{\mathrm{mm}}\|} \right\rangle = 1$.*

The proof sketch below illustrates how the NTP-separability/compatibility assumptions influence the outcome and why the regularization path induces an optimization bias toward the NTP-SVM direction. Complementing Thm. 1, we also show (see Lemma 4 in the appendix) that $\lim_{B \to \infty} \mathcal{P}_{\mathcal{F}}(\boldsymbol{W}_B) = \boldsymbol{W}^\star$. These together provide a complete characterization of the implicit optimization bias of (8).

---

[7]The necessity for such large $d$ can be mitigated through the utilization of non-linear architectures (such as an MLP decoder), in which the total number of parameters can be increased by augmenting the width or depth, rather than directly modifying the embedding dimension $d$ as in linear models.

*Proof sketch (App. E.2 for details).* We first show $\widehat{\boldsymbol{W}}_B$ is on the boundary: $\|\widehat{\boldsymbol{W}}_B\| = B$. If not, then $\langle \nabla \mathrm{CE}(\widehat{\boldsymbol{W}}_B), \boldsymbol{W}^{\mathrm{mm}}\rangle = 0$. But, few algebraic manipulations show $\langle -\nabla \mathrm{CE}(\widehat{\boldsymbol{W}}_B), \boldsymbol{W}^{\mathrm{mm}}\rangle$ equals

$$\sum_{j\in[m]} \hat{\pi}_j \sum_{z\in\mathcal{S}_j} \hat{p}_{j,z}\Big( \sum_{z'\in\mathcal{S}_j, z'\neq z} s_{j,z'}\,(\boldsymbol{e}_z - \boldsymbol{e}_{z'})^{\top}\boldsymbol{W}^{\mathrm{mm}}\bar{\boldsymbol{h}}_j + \sum_{v\notin\mathcal{S}_j} s_{j,v}\,(\boldsymbol{e}_z - \boldsymbol{e}_v)^{\top}\boldsymbol{W}^{\mathrm{mm}}\bar{\boldsymbol{h}}_j \Big),$$

where we denote $s_{j,v} \coloneqq \mathbb{S}_v(\widehat{\boldsymbol{W}}_B\bar{\boldsymbol{h}}_j) > 0, v \in \mathcal{V}, j \in [m]$. The first term in the parenthesis is zero by (6a), while the second term is strictly positive by (6b), leading to contradiction.

Now, consider a 'genie' point $\boldsymbol{W}_B^{\star} = \boldsymbol{W}^{\star} + R(B)\cdot\boldsymbol{W}^{\mathrm{mm}}$, where $\boldsymbol{W}^{\star} \in \mathcal{F}$ satisfies (4), and $R = R(B)$ is chosen such that $\|\boldsymbol{W}_B^{\star}\| = B$. We will show that $\boldsymbol{W}_B^{\star}$ attains a small CE loss as $B$ (hence, $R$) grows. To do this, denote for convenience the logits

$$\ell_{j,v}^{\star} \coloneqq \boldsymbol{e}_v^{\top}\boldsymbol{W}^{\star}\bar{\boldsymbol{h}}_j \quad \text{and} \quad \ell_{j,v}^{\mathrm{mm}} \coloneqq \boldsymbol{e}_v^{\top}\boldsymbol{W}^{\mathrm{mm}}\bar{\boldsymbol{h}}_j$$

for all for $v \in \mathcal{V}, j \in [m]$, and note that $\boldsymbol{e}_v^{\top}\boldsymbol{W}_B^{\star}\bar{\boldsymbol{h}}_j = \ell_{j,v}^{\star} + R\,\ell_{j,v}^{\mathrm{mm}}$. By using (4) and (6a):

$$\sum_{z'\in\mathcal{S}_j} e^{-(\ell_{j,z}^{\star}+R\ell_{j,z}^{\mathrm{mm}}-\ell_{j,z'}^{\star}-R\ell_{j,z'}^{\mathrm{mm}})} = \sum_{z'\in\mathcal{S}_j} e^{-(\ell_{j,z}^{\star}-\ell_{j,z'}^{\star})} = \sum_{z'\in\mathcal{S}_j} \frac{\hat{p}_{j,z'}}{\hat{p}_{j,z}} = \frac{1}{\hat{p}_{j,z}}.$$

Moreover, using (6b) and defining $C \coloneqq V e^{\|\boldsymbol{W}^{\star}\|M}$ for $M \coloneqq \sqrt{2}\cdot\max_{j\in[m]}\|\bar{\boldsymbol{h}}_j\|$, gives:

$$\sum_{v\notin\mathcal{S}_j} e^{-(\ell_{j,z}^{\star}+R\ell_{j,z}^{\mathrm{mm}}-\ell_{j,v}^{\star}-R\ell_{j,v}^{\mathrm{mm}})} \leq e^{-R}\sum_{v\notin\mathcal{S}_j} e^{-(\ell_{j,z}^{\star}-\ell_{j,v}^{\star})} \leq C\,e^{-R}.$$

Combining the above within Eq. (2), using $\log(1 + x) \leq x, x > 0$ and the fact that $\hat{\pi}_j, \hat{p}_{j,z}$ are probabilities, yields:

$$\mathrm{CE}(\boldsymbol{W}_B^{\star}) \leq \sum_{j\in[m]} \hat{\pi}_j \sum_{z\in\mathcal{S}_j} \hat{p}_{j,z} \log\Big( \frac{1}{\hat{p}_{j,z}} + C\,e^{-R}\Big) \leq \mathcal{H} + C\,e^{-R}. \tag{9}$$

Next, towards contradiction, we will show that if $\widehat{\boldsymbol{W}}_B$ is *not* in the direction of $\boldsymbol{W}^{\mathrm{mm}}$, then it incurs a loss that is larger than $\mathrm{CE}(\boldsymbol{W}_B^{\star})$. The trick here is to bound the KL divergence term:

$$\mathrm{CE}(\widehat{\boldsymbol{W}}_B) - \mathcal{H} = \sum_{j\in[m]} \hat{\pi}_j \sum_{z\in\mathcal{S}_j} \hat{p}_{j,z} \log\Big( \hat{p}_{j,z}\big( \sum_{z'\in\mathcal{S}_j} e^{\ell_{j,z'}-\ell_{j,z}} + \sum_{v\notin\mathcal{S}_j} e^{\ell_{j,v}-\ell_{j,z}}\big)\Big), \tag{10}$$

where we denote logits $\ell_{j,v} \coloneqq \boldsymbol{e}_v^{\top}\widehat{\boldsymbol{W}}_B\bar{\boldsymbol{h}}_j$. Assume there exists $\epsilon > 0$ and arbitrarily large $B$ satisfying:

$$\Big\|\big(\|\boldsymbol{W}^{\mathrm{mm}}\|/B\big)\,\widehat{\boldsymbol{W}}_B - \boldsymbol{W}^{\mathrm{mm}}\Big\| > \epsilon. \tag{11}$$

Define $\widehat{\boldsymbol{W}} = (\widehat{\boldsymbol{W}}_B - \boldsymbol{W}^{\star})/R'(B)$, where $R' = R'(B) > 0$ can be chosen so that $\|\widehat{\boldsymbol{W}}\| = \|\boldsymbol{W}^{\mathrm{mm}}\|$. Further choose $B$ large enough so that Eq. (11) guarantees $\|\widehat{\boldsymbol{W}} - \boldsymbol{W}^{\mathrm{mm}}\| \geq \epsilon'$, for some $\epsilon' > 0$. Since $\boldsymbol{W}^{\mathrm{mm}}$ is the unique minimizer of (NTP-SVM) and $\|\widehat{\boldsymbol{W}}\| = \|\boldsymbol{W}^{\mathrm{mm}}\|$, there exists $\delta \in (0,1)$ and $j \in [m]$ such that at least one of the following is true: *(i)* $\exists z$ and $z' \neq z \in \mathcal{S}_j$ such that $|(\boldsymbol{e}_z - \boldsymbol{e}_{z'})^{\top}\widehat{\boldsymbol{W}}\bar{\boldsymbol{h}}_j| \geq \delta$ *(ii)* $\exists z \in \mathcal{S}_j, v \notin \mathcal{S}_j$ such that $(\boldsymbol{e}_z - \boldsymbol{e}_v)^{\top}\widehat{\boldsymbol{W}}\bar{\boldsymbol{h}}_j \leq 1 - \delta$.

*Case (i):* Without loss of generality $(\boldsymbol{e}_z - \boldsymbol{e}_{z'})^{\top}\widehat{\boldsymbol{W}}\bar{\boldsymbol{h}}_j \leq -\delta$ (otherwise, flip $z, z'$). Thus, ignoring all but the $(j, z, z')$-term in (10) and using $\ell_{j,z'} - \ell_{j,z} \geq R'\delta + \log\big(\frac{\hat{p}_{j,z'}}{\hat{p}_{j,z}}\big)$ gives

$$\mathrm{CE}(\widehat{\boldsymbol{W}}_B) - \mathcal{H} \geq \hat{\pi}_j\hat{p}_{j,z} \log\Big( \hat{p}_{j,z}e^{(\ell_{j,z'}-\ell_{j,z})}\Big) \geq \frac{1}{n}\log\Big(\frac{e^{R'\delta}}{n}\Big).$$

Comparing this to (9) for large enough $B$ gives that $\mathrm{CE}(\widehat{\boldsymbol{W}}_B) > \mathrm{CE}(\boldsymbol{W}_B^{\star})$, a contradiction.

*Case (ii):* We can assume $\widehat{\boldsymbol{W}} \in \mathcal{F}^{\perp}$, since otherwise we are in Case (i). Now, again ignoring all but the $(j, z)$ term in the CE loss for which the assumption holds for some $v \notin \mathcal{S}_j$, we find

$$\mathrm{CE}(\widehat{\boldsymbol{W}}_B) - \mathcal{H} \geq \hat{\pi}_j\hat{p}_{j,z} \log\Big( \hat{p}_{j,z}\big( \sum_{z'\in\mathcal{S}_j} e^{(\ell_{j,z'}-\ell_{j,z})} + e^{(\ell_{j,v}-\ell_{j,z})}\big)\Big).$$

Using $\mathcal{P}_{\mathcal{F}}(\widehat{\boldsymbol{W}}_B) = \boldsymbol{W}^\star$ and (4) yields $\sum_{z' \in \mathcal{S}_j} e^{(\ell_{j,z'} - \ell_{j,z})} = \frac{1}{\hat{p}_{j,z}}$ . Moreover, by assumption of Case (ii): $e^{\ell_{j,v} - \ell_{j,z}} \geq e^{-R'(1-\delta)} e^{\ell_{j,v}^\star - \ell_{j,z}^\star} \geq c' e^{-R'(1-\delta)}$, for $c' := e^{-\|\boldsymbol{W}^\star\|M}$. Putting together yields:

$$\mathrm{CE}(\widehat{\boldsymbol{W}}_B) - \mathcal{H} \geq \hat{\pi}_j \hat{p}_{j,z} \log\left(1 + \hat{p}_{j,z} c' e^{-R'(1-\delta)}\right) \geq c' e^{-R'(1-\delta)}/2n^2 ,$$

where the second inequality uses $\log(1 + x) \geq \frac{x}{1+x}, x > 0$. Compare this with (9): For large enough $B$, since $R, R'$ grow at the same rate, it holds $\frac{c'}{2n^2} e^{-R'(1-\delta)} > C e^{-R}$. Thus, $\mathrm{CE}(\widehat{\boldsymbol{W}}_B) > \mathrm{CE}(\boldsymbol{W}_B^\star)$, a contradiction. In either case, we arrive at a contradiction, which completes the proof. $\qquad\square$

## 5   Gradient Descent

This section studies the implicit bias of GD. Denote the GD iterates at time $k$ by $\boldsymbol{W}_k = \boldsymbol{W}_{k-1} - \eta \nabla \mathrm{CE}(\boldsymbol{W}_{k-1})$ for arbitrary initial point $\boldsymbol{W}_0$ and constant step-size $\eta > 0$ small enough to guarantee descent. The first observation is that the norm of the GD iterates increases with iterations.

**Lemma 2** (Norm growth). *If training data are NTP$_{\mathcal{H}}$-compatible and NTP-separable, then* $\lim_{k \to \infty} \mathrm{CE}(\boldsymbol{W}_k) = \mathcal{H}$ *and* $\lim_{k \to \infty} \|\boldsymbol{W}_k\| = \infty$.

This is intuitive because the CE loss is convex in $\boldsymbol{W}$ (thus, GD approaches the objective's infimum $\mathcal{H}$), and, in view of Proposition 1, the CE loss at all finite $\boldsymbol{W}$ is bounded away from $\mathcal{H}$. The relevant question then becomes that of determining the limit of the direction of the GD iterates.

**Theorem 2** (Implicit bias of GD). *Assume NTP$_{\mathcal{H}}$-compatible and NTP-separable training data $\mathcal{T}_m$. Then, it holds that* $\lim_{k \to \infty} \left\langle \frac{\boldsymbol{W}_k}{\|\boldsymbol{W}_k\|}, \frac{\boldsymbol{W}^{\mathrm{mm}}}{\|\boldsymbol{W}^{\mathrm{mm}}\|} \right\rangle = 1$ . *Moreover,* $\lim_{k \to \infty} \mathcal{P}_{\mathcal{F}}(\boldsymbol{W}_k) = \boldsymbol{W}^\star$.

The theorem establishes [8] that in the limit of iterations: $\boldsymbol{W}_k \approx \boldsymbol{W}^\star + \|\mathcal{P}_\perp(\boldsymbol{W}_k)\| \overline{\boldsymbol{W}^{\mathrm{mm}}}$, which is analogous to the result we obtained previously for the regularization path. Although its proof is more involved compared to the proof of Thm. 1, the proof of its main ingredient (Lem. 5 in the appendix) is conceptually similar: It involves comparing the loss $\mathrm{CE}(\boldsymbol{W}_k)$ for large iterations $k$ to the loss evaluated at a "genie" point that is chosen so that: (i) On the subspace $\mathcal{F}$, it agrees with $\boldsymbol{W}_k$. This is because it is easy to show that $\mathcal{P}_{\mathcal{F}}(\boldsymbol{W}_k)$ converges to $\boldsymbol{W}^\star$ by standard gradient descent analysis for convex functions; (ii) On the orthogonal subspace $\mathcal{F}^\perp$, it follows the optimal (with respect to accelerating loss decrease) max-margin direction $\overline{\boldsymbol{W}^{\mathrm{mm}}} \in \mathcal{F}^\perp$. To establish the loss comparison, the ideas is to compare the values of the adjusted loss $\mathrm{CE}_\perp(\boldsymbol{W}) := \mathrm{CE}(\boldsymbol{W}) - \mathrm{CE}(\mathcal{P}_{\mathcal{F}}(\boldsymbol{W}))$.

We validate our analysis with experiments on synthetic data in App. A. For illustration, Fig. 1 shows a 2D setting with $m = 3$ distinct contexts, each followed by $S_j = 3$ tokens/words out of total $V = 5$ words in the vocabulary. The left subfigure illustrates: (i) In black markers, the context-embedding geometry along with the associated support sets for each context A, B, and C. (ii) In colored markers, the geometry of word-embeddings, that is the max-NTP-margin vectors $(\boldsymbol{W}^{\mathrm{mm}})^\top \boldsymbol{e}_v, v \in [5]$, to which GD directionally converges. See caption for interpretation and Fig. 2 in the App. for vis. of the finite component of word-embeddings on the subspace $\mathcal{F}$. The right subfigure shows results of GD training with respect to training loss, norm growth, alignment with $\boldsymbol{W}^{\mathrm{mm}}$, and convergence to $\boldsymbol{W}^\star$ on $\mathcal{F}$. See App. A for further implementation details and additional experiments.

## 6   Related work

We build on the literature on implicit optimization bias of CE loss in one-hot supervised classification. [76] show that for linear models and linearly-separable data, GD converges in direction to the max-margin classifier. This result strengthens [68] that showed the regularization path of CE minimization converges to the same limit. Closer to us, [34, 37] extend the analysis to encompass general binary data as follows: the data are linearly separable only on a certain subspace, and they show that GD converges, in direction, towards the max-margin classifier confined within that subspace. On the orthogonal subspace, it converges to a finite point. While operationally similar, Thms. 1, 2 cannot

---

[8]In line with observations in one-hot encoding [59], we anticipate the directional behavior remains unchanged under stochasticity, e.g. when using SGD to minimize (2). Yet, note a subtle but crucial difference in applying SGD to (1) vs (2), as the latter involves sampling *distinct* contexts in each iteration. In this latter case, we also point out that favorable interpolation conditions, such as strong-growth (e.g., [91]), can be shown to hold.

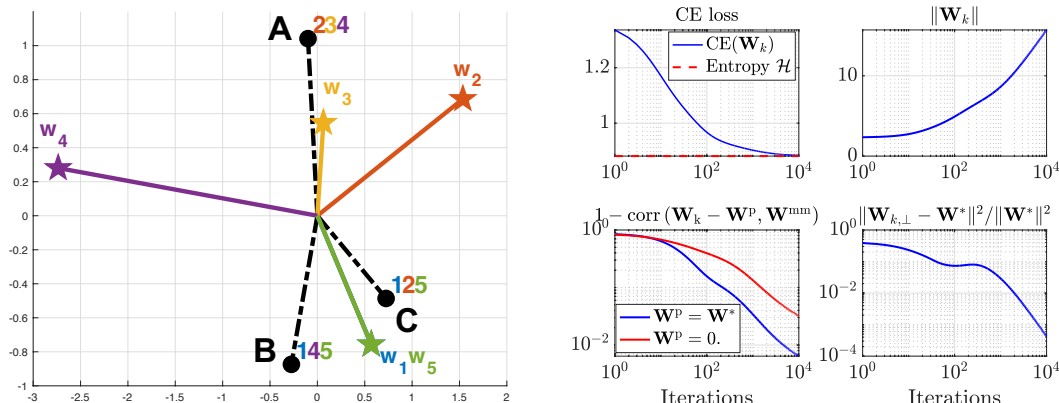

Figure 1: Vis. of NTP implicit optimization bias in a setting with $m = 3$ *distinct* contexts, embedding dimension $d = 2$, vocabulary of $|\mathcal{V}| = 5$ words and support sets of length $|\mathcal{S}_j| = 3, j \in [3]$. ***Left:*** Vis. of *context embeddings* $\bar{\boldsymbol{h}}_j$ in circle black markers (marked as A,B,C) and of their associated support sets $\mathcal{S}_j$ (colored text below each marker). Colored vectors (star markers) represent max-NTP-margin vectors $\boldsymbol{w}_v^\top := \boldsymbol{e}_v^\top \boldsymbol{W}^{\mathrm{mm}}, v \in [5]$ found by GD. Interpreting decoder vectors as *word embeddings* leads to intuitive findings on their geometry learned by NTP training. E.g., word embedding $\boldsymbol{w}_3$ (almost) aligns with context-embedding $A$ and the normal hyperplane it defines separates $A$ from $B$ and $C$, since word 3 only appears after context $A$. The rest of the words follow two contexts each and their word-representation naturally belongs to the cone defined by the embeddings of those respective contexts. The wider the cone, the larger the magnitude of the word embedding to compensate for the large angle between context-representations that share the same next-word. Note that geometry of depicted word embeddings only depends on support sets, but the conditional probabilities define another set of word representations on an orthogonal (matrix) subspace; see text for details and vis. ***Right:*** Upper/lower graphs confirm the predictions of Lemma 2 and of Theorem 2, respectively.

be directly derived from theirs since our setting is neither binary nor one-hot. Nevertheless, our proofs extend the foundational work of [68, 34, 37], akin to numerous other studies that explore extensions to nonlinear architectures[50, 35, 28, 29, 83, 89], and to stochastic and adaptive algorithms [60, 64, 21, 47, 77, 3, 14, 2]. The implicit bias viewpoint has also created opportunities to study generalization in overparameterized settings. [31, 4, 57, 22] build a two-stage approach initially leveraging implicit bias to simplify the complexities of optimization before addressing generalization. This narrows the generalization question to the properties of the corresponding max-margin classifier [58, 13, 43, 78, 23, 100, 72, 94]. The same strategy has also been adopted to study model robustness to adversarial perturbations [33, 80, 16], out-of-distribution data [87], and imbalances [69, 15, 42]. Our results motivate such extensions in the richer NTP setting.

Recent work [49] also studies forms of implicit bias for language models trained to reach the risk lower bound. However, they assume training with population loss and analyze implicit bias through Hessian-trace minimization without providing explicit parameter characterizations as in Thm. 2. Crucially, their results do *not* apply to CE loss[9] or to sparse support-sets. Another interesting work [52] studies learning abilities of autoregressive training and inference. However, their findings do *not* apply to NTP as they inherently assume each context is followed by a unique next token.

Finally, although stemming from different perspectives, the form of our convergence results echoes a recent conjecture by [82] regarding implicit optimization bias in transformers. Unlike their conjecture, which focuses on binary classification, our results are rigorously proven and apply to the NTP setting. Further detailed discussion on related follow-up work on implicit optimization bias in self-attention architectures, as initiated by [83], is deferred to Appendix B. In contrast to this line of work, we here focus on the optimization biases of the NTP training-paradigm itself, which is orthogonal to the intricacies of the specific architecture generating the context embeddings.

---

[9][49, Thm. 4.3] uses [47, Cor. 5.2], which applies to regression on scalar labels; thus is not applicable in NTP.

# 7 Conclusion, limitations and future work

Towards characterizing implicit regularization effects, we highlight two key aspects of NTP training: *(i)* Formulating it as CE optimization over *distinct* contexts; this is long recognized in language modeling (e.g., [44, 63]) since Shannon's initial work, yet seemingly overlooked in recent studies, such as [49, 52]. *(ii)* Accounting for *sparsity* in the matrix of next-token conditional probabilities. While traditional language modeling techniques often mitigate sparsity using smoothing heuristics that assign non-zero probabilities to unobserved next tokens [44, 63, 39], we recognize sparsity as a critical factor in NTP optimization that influences parameter divergence[10].

As the first study of implicit biases in NTP training, our results are based on several assumptions essential for establishing an initial foundational understanding. The framework allows for various exciting promising research directions, some of which we outline below.

Even within the assumed linear setting and GD, interesting directions involve:

• **NTP-separability thresholds:** Identifying exact thresholds for NTP-separability under distributional assumptions, akin to previous work on one-hot separability (Remark 1). However, relaxing the overparameterization requirement that the embedding dimension $d$ be proportional to the number of distinct contexts $m$ would necessitate exploring non-convex architectures (see 'Memory capacity' below).

• **Generalization:** Studying generalization in NTP settings by examining statistical properties of the NTP-SVM solution. Past research has successfully undertaken similar investigations for one-hot classification (see Sec. 6). While we acknowledge the importance of addressing specific challenges inherent to NTP —such as determining an appropriate measure of generalization, or establishing suitable statistical models for context-embeddings that respect the discrete nature of the underlying token subsequences—we believe this direction holds promise for further exploration.

In addition to these, essential extensions include relaxing the linearity assumption.

• **Architecture-specific embeddings:** A bottom-up approach considering architecture-specific embeddings could begin by modeling the embeddings produced by, for instance, a shallow transformer and analyzing the effects of optimization biases on the training of both the transformer and the decoder weights. This complements the works of [83, 82], who investigate one-layer self-attention with a fixed decoder. A challenge in this approach is balancing the restriction to shallow transformers (for analytical tractability) with ensuring that the NTP loss reaches the entropy lower bound. This may require constraining the training data distribution, for example, to a Markov chain [51, 25].

• **Memory capacity in NTP settings:** Without imposing further restrictions on the data beyond the discrete nature of tokens from a finite vocabulary, there is a strong case for investigating the memory capacity of sequence-to-sequence architectures, such as transformers, in the context of NTP. Recent studies on transformer memory capacity [40, 41] do *not* apply here.

• **Unconstrained features:** Extending the top-down approach, one could consider freely optimizing context embeddings together with decoder vectors (also known as word embeddings). The resulting log-bilinear model, reminiscent of wor2vec models [63, 55], extends the unconstrained features model, which has recently been employed to investigate neural collapse geometry in one-hot classification settings [56]. This idea offers a promising avenue for uncovering structures in the geometries of context and word embeddings when learned jointly, potentially revealing new insights into the capabilities of sufficiently expressive language models (see Fig. 1 for cases involving only the latter).

• **Other optimizers:** Exploring the NTP implicit bias of adaptive algorithms, such as Adam, potentially building on recent works in this area focused on one-hot classification [96, 95].

We hope this work inspires further research in the discussed directions, contributing to a deeper understanding of the intricacies involved and potentially yielding improvements in NTP training.

**Acknowledgements**

Thank you to Tina Behnia, Yize Zhao, Vala Vakilian, and Puneesh Deora for inspiring discussions that contributed to this work and for their valuable suggestions on the manuscript. I am also grateful to Gautam Goel for his careful reading and for pointing out several typos. Thanks to the anonymous reviewers for their feedback. This work is supported by the NSERC Discovery Grant No. 2021-03677, the Alliance Grant ALLRP 581098-22, NFRFE-2023-00936, and a CIFAR AI Catalyst Grant.

---

[10]Parameter divergence in transformer-based language models has been empirically observed in [53].

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

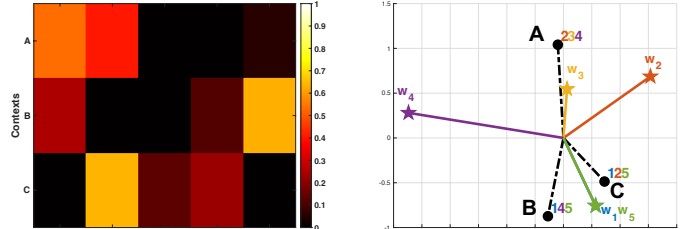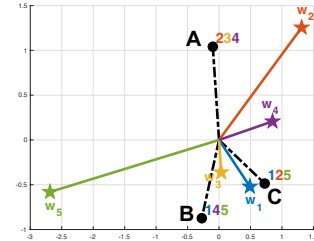

Figure 2: Same setup as Fig. 1. ***Left:*** Matrix $\boldsymbol{P}$ of conditional probabilities of words (cols.) per context (rows). Each row corresponds to the conditional probability vectors $\boldsymbol{p}_j, j \in [m]$. Black entries correspond to off-support words. ***Middle:*** Shown as $\boldsymbol{w}_z, z \in [5]$, the rows of the NTP-SVM solution $\boldsymbol{W}^{\mathrm{mm}}$ to which GD directionally converges. ***Right:*** Shown as $\boldsymbol{w}_z, z \in [5]$, the rows of the finite parameter $\boldsymbol{W}^\star$ to which GD iterates projected on $\mathcal{F}$ converge to. The geometry of $\boldsymbol{W}^{\mathrm{mm}}$ depends only on the support-set of $\boldsymbol{P}$. On the other hand, the geometry of $\boldsymbol{W}^\star$ depends on the entries of $\boldsymbol{P}$ for in-support tokens/words. As seen from visualization of $\boldsymbol{P}$, the words 1 and 5 have the same support pattern (i.e., both follow the same contexts $A$ and $B$). Thus, $\boldsymbol{w}_1 = \boldsymbol{w}_5$ in the Middle plot. However, on the subspace $\mathcal{F}$ corresponding to the Right plot, $\boldsymbol{w}_1 \neq \boldsymbol{w}_5$, which allows matching the different conditional probabilities with which each follows contexts $A$ and $B$.

## A   Experiments

All experiments were conducted on a MacBook Pro equipped with a 2.3 GHz Quad-Core Intel Core i7 processor and 32 GB of memory. The experiments are of relatively small scale and were implemented in Matlab. The code is straightforward to reproduce, following the detailed specifications provided in the subsequent sections. For completeness, the code will be made publicly available on Github in the final version of the paper.

### A.1   Additional details on 2D example of Fig. 1

Figure 1 illustrates a toy 2d example where the embeddings and the hyperplanes defined by each row of $\boldsymbol{W}^{\mathrm{mm}}$ can be visualized. We used $d = 2, m = 3, V = 5$ and $S_1 = S_2 = S_3 = 3$. The support sets of each embedding are shown in the figure color-coded to match the respective decoder hyperplane. Probabilities are assigned randomly. The empirical conditional entropy evaluates to $\mathcal{H} = 0.8811$ and the matrix of conditional probabilities is visualized in Figure 2. In the same figure, we also visualize the rows of the directional component $\boldsymbol{W}^{\mathrm{mm}}$ (Middle) and of the finite component $\boldsymbol{W}^\star$ (Right). Interpreting the $V \times d$ decoder matrix as the matrix of learned word embeddings, this provides a visualization of their geometry. As per our results, the two word-embedding matrices $\boldsymbol{W}^\star$ and $\boldsymbol{W}^{\mathrm{mm}}$ lie on orthogonal subspaces. The geometry of the first depends on the probabilities of in-support tokens, while that of the second depends only on the support set of these probabilities. See also caption of Fig. 2.

### A.2   Overparameterized setting

We examine the implicit bias of GD on NTP training with overparameterization on synthetic data generated as follows. We construct dataset with $n = 5000$ sequences involving $m = 50$ distinct contexts. Each distinct context gets mapped to a randomly generated embedding of dimension $d = 60 > m$. We set vocabulary size $V = 10$ and each context $j \in [m]$ is followed by $S_j = 6, \forall j \in [m]$ possible next-tokens. The support sets $\mathcal{S}_j \subset \mathcal{V}$ and the probabilities $\hat{p}_{j,z}, z \in \mathcal{S}_j$ are chosen randomly; see Fig. 3 for representative examples from the training dataset. For a fixed realization of the dataset (for which $\mathcal{H} \approx 1.445$nats), we run GD, normalized GD (NGD), and Adam from random LeCun initialization. For GD, we use learning rate $\eta = 0.5$ and for NGD and Adam $\eta = 0.01$. For Adam, we also set $\beta_1 = 0.9, \beta_2 = 0.99$. We run all algorithms for $1e4$ iterations. For each case, we plot the following as a function of iterations:

1. Upper Left: CE loss versus entropy lower bound
2. Upper Right: parameter norm growth

3. Lower Left: correlation of $\boldsymbol{W}^{\mathrm{mm}}$ with iterates $\boldsymbol{W}_k$ and of "corrected" iterates $\boldsymbol{W}_k - \boldsymbol{W}^\star$ after substracting the component on $\mathcal{H}$

4. Lower Right: convergence of the subspace component $\boldsymbol{W}_{k,\mathcal{F}} = \mathcal{P}_{\mathcal{F}}(\boldsymbol{W}_k)$.

Fig. 4 shows an instance of these. As predicted by our analysis, in this overparameterized setting: CE loss converges to its lower-bound, parameter norm increases, iterates align in direction with $\boldsymbol{W}^{\mathrm{mm}}$, and the subspace component converges to $\boldsymbol{W}^\star$.

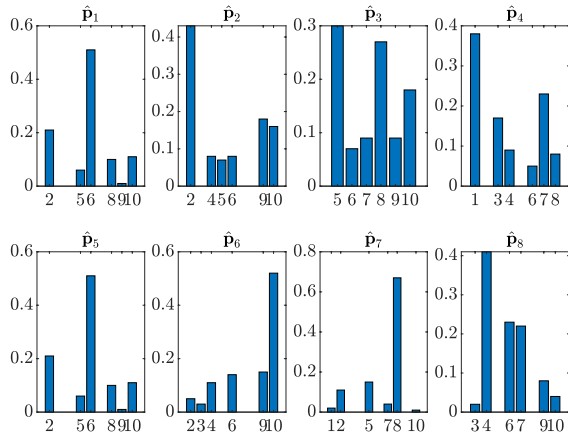

Figure 3: Eight randomly picked contexts with their associated next-token empirical conditional probabilities $\hat{\boldsymbol{p}}_j$. The indices shown on the x-axis define the support set $\mathcal{S}_j$ of each context.

Figure 5 illustrates the same plots, but this time for training over the same dataset with NGD and Adam. We observe same implicit bias, but faster convergence. For NGD, this is consistent with analogous findings (rigorous in that case) for one-hot classification [60, 36].

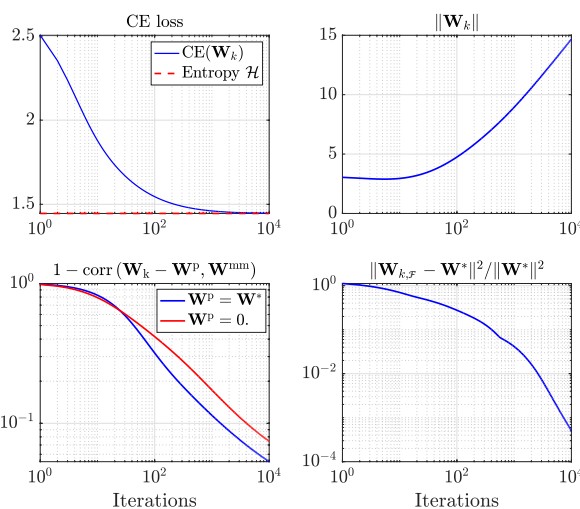

Figure 4: Experimental illustration of the implicit bias of GD in NTP over synthetic data with overparameterization. See App. A for detailed description of the experimental setting. The upper two graphs confirm the predictions of Lemma 2, while the lower two graphs adhere to the predictions of Theorem 2.

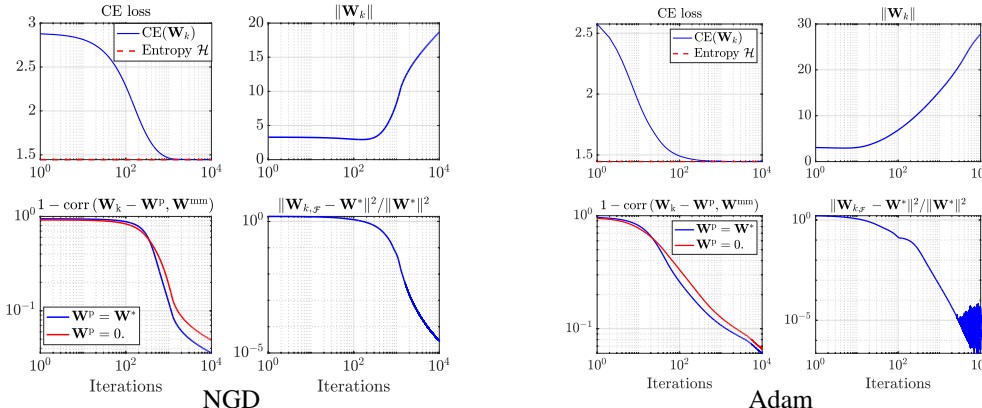

Figure 5: Implicit bias of *normalized* GD (Left) and of Adam (Right) in NTP over synthetic data with overparameterization. Both exhibit the same implicit bias, but converge faster than GD, with Adam being slightly faster than NGD.

# B  Additional related work

**Implicit bias in transformers.** As already mentioned in Sec. 6, our work is closely related to [82], where the authors investigate the implicit bias of self-attention in transformers. The insight put forth in the prequel [83] is that softmax attention induces implicit-bias behaviors that bear similarities to vanilla implicit bias of one-hot prediction. Concretely, [82] studies GD optimization of one-layer self-attention with fixed decoder and *one-hot binary* classification. They show that, in the limit, GD finds attention weights that converge in direction to the solution of an SVM problem that separates optimal tokens from non-optimal ones. Their non-convex setting introduces locally optimal SVM directions to which GD may converge depending on initialization. Different to them, the NTP setting that we study involves predictions over multiple categories and is *not* one-hot. Also, while they fix the decoder, here, we fix the embeddings. In these respects their results are rather different. More similarities arise when [82] replace the linear decoder with a MLP, which they note can induce multiple optimal tokens per sequence. This leads them to formulate a more general token-separating SVM program, which similar to ours confines the separation on a certain data subspace. However, the operational nature of the programs remains different as theirs optimizes attention weights and separates tokens within a sequence, while ours optimizes decoder weights and separates context embeddings based on their respective support sets. More importantly, while [82] only conjectures the convergence of GD to their general SVM program, we leverage convexity in our setting to prove an analogous statement rigorously. Eventually, as we move lower in our top-down approach and consider architecture-specific embeddings generated by attention, we anticipate to see integration of our ideas with theirs.

Beyond [82], there is growing recent research investigating optimization and generalization principles of transformers, e.g., [70, 24, 48, 93, 99, 1, 45, 83, 82, 84, 17]. These efforts predominantly employ a 'bottom-up' approach that involves isolating shallow transformers, often with simplifications such as removing MLPs, utilizing single heads instead of multiple, and fixing certain parts while training only a subset of trainable parameters. Most of these studies have focused on classical one-hot supervised settings, and only a handful (e.g., [84, 85]) have sought extending these 'bottom-up' analyses to NTP settings. Yet, their primary emphasis remains on uncovering the role of attention and how attention weights evolve during training. Instead, our approach uniquely emphasizes the NTP training paradigm itself, shifting the focus from the intricacies of specific transformer architectures.

Upon completing this paper, we became aware of independent contemporaneous research by Li et al. [46] that also examines the implicit bias of self-attention with a fixed linear decoder in next-token prediction scenarios. Unlike our study which utilizes the widely adopted CE loss, their approach is based on log-loss, which renders the training loss convex, a similarity shared with our model despite the inclusion of self-attention. Both our results and those of Li et al. substantiate the conjecture posited by Tarzanagh and colleagues [82], albeit in very distinct settings. Notably, contrary to both

[83] and [46], we unveil the optimization intricacies of the NTP paradigm, even within the simplest linear settings.

**Classification with soft labels.** Unlike one-hot classification, soft-label classification associates each example with a probability vector, where each entry represents the likelihood of a corresponding label characterizing the example. Although arguably less prevalent than one-hot (or hard-label) classification, soft-label classification arises in various contexts, including modeling human confusion during crowd-sourcing [65, 75, 18], knowledge distillation [32], label smoothing [79], and mixup [98]. Our model of last-token prediction also falls within this setting. Specifically, our approach is most closely related to soft-labels generated by averaging annotators' hard labels [65], rather than following the winner-takes-all rule to assign labels. [65] and follow-up work have provided empirical evidence that using probabilistic soft labels generated from crowd annotations for training leads to improved performance in terms of model generalization, calibration, and robustness to out-of-distribution data. To the best of our knowledge, no prior work has investigated the implicit bias of gradient descent in this or other soft-label classification settings; thus, our results are of direct relevance to these contexts as well.

## C  Autoregressive setting

For concreteness and simplified notation, in the paper's main body we focus on NTP over sequences of fixed length. We show here that this encompasses the autoregressive (i.e., sequential) setting with minimal changes. This also emphasizes the role played in our results by the sequence length.

As pointed in (1), the full autoregressive NTP objective averages $T$ individual losses (without loss of generality assume sequences of equal maximum length $T$). In order to make our analysis applicable, we first need to express (1) in terms of *unique* contexts. Mirroring the notations in Sec. 2, define the following for $t \in [T - 1]$:

- $m_t, t \in [T-1]$ is the number of *distinct* contexts of size $t$. Note that $m_1 \geq m_2 \geq \cdots \geq m_{T-1}$.
- $m = \sum_{t=1}^{T-1} m_t$ is the total number of distinct contexts in the dataset
- $\bar{h}_{t,j} := h_\theta(\bar{x}_{j,t}), t \in [T-1], j \in [m_t]$ is the embedding of the $j$-th (among all $t$-long contexts) distinct context $\bar{x}_{j,t}$.
- $\hat{\pi}_{j,t}$ is the empirical probability of $\bar{x}_{j,t}$.
- $\hat{p}_{j,t,z}$ is the empirical probability that context $\bar{x}_{j,t}$ is followed by token $z \in \mathcal{V}$.
- $\mathcal{S}_{j,t}$ is the support set of the next-token distribution of context $\bar{x}_{j,t}$.

With this notation, the NTP objective becomes

$$\text{CE} = - \sum_{t \in [T-1]} \sum_{j \in [m_t]} \hat{\pi}_{t,j} \sum_{z \in \mathcal{S}_{j,t}} \hat{p}_{t,j,z} \log \left( \mathbb{S}_z(W \bar{h}_{t,j}) \right).$$

To continue enumerate the multi-set $\mathcal{I} := \{i = (j, t) \,|\, t \in [T-1], j \in [m_t]\}$. We may then rewrite the above as

$$\text{CE} = - \sum_{i \in \mathcal{I}} \hat{\pi}_i \sum_{z \in \mathcal{S}_i} \hat{p}_{i,z} \log \left( \mathbb{S}_z(W \bar{h}_i) \right).$$

At this point note that this is of identical form to (2). Consequently, the definitions (e.g., NTP-separability, NTP-margin) and results derived in the main body for sequences of fixed length are applicable to the AR setting, extending mutatis mutandis.

**Remark 2** (The role of sequence length.)**.** *Despite the above reduction of the AR setting to the fixed-length setting, it is crucial to recognize that sequence length remains a significant factor in the AR model. Specifically, it influences the formulation through support sets and their associated probabilities. As sequences extend in length, their corresponding support sets generally become sparser, indicative of less ambiguity in predicting the next token. This dynamic is captured by Shannon's inequality,*

$$\mathcal{H}_t \geq \mathcal{H}_{t+1}, \text{ where } \mathcal{H}_t = - \sum_{j \in [m_t]} \sum_{z \in \mathcal{S}_{t,j}^\ell} \pi_{t,j} \hat{p}_{t,j,z} \log(\hat{p}_{t,j,z}),$$

*reflecting the incremental reduction in entropy as sequence length increases.*

# D   Notations

Throughout, lowercase and uppercase bold letters (e.g., $\boldsymbol{a}$ and $\boldsymbol{A}$) represent vectors and matrices, respectively. $\langle \cdot, \cdot \rangle$ and $\|\cdot\|$ denote Euclidean inner product and norm, respectively. For matrix $\boldsymbol{A}$, we denote its pseudoinverse as $\boldsymbol{A}^{\dagger}$. All logarithms are natural logarithms (base $e$). We denote $\boldsymbol{e}_v$ the $v$-th standard basis vector in $\mathbb{R}^V$. $\Delta^{V-1}$ denotes the $V$-dimensional unit simplex and $\mathbb{S}() : \mathbb{R}^V \to \Delta^{V-1}$ the softmax map:

$$\mathbb{S}(\boldsymbol{a}) = [\mathbb{S}_1(\boldsymbol{a}), \ldots, \mathbb{S}_V(\boldsymbol{a})]^{\top}, \qquad \text{with} \ \ \mathbb{S}_v(\boldsymbol{a}) = \frac{e^{\boldsymbol{e}_v^{\top} \boldsymbol{a}}}{\sum_{v' \in [V]} e^{\boldsymbol{e}_{v'}^{\top} \boldsymbol{a}}} \, .$$

As explained in Section 2 we represent a training set as

$$\mathcal{T}_m := \{(\bar{\boldsymbol{h}}_j, \hat{\pi}_j, \hat{p}_{j,z \in \mathcal{V}})\}_{j \in [m]} \, .$$

We assume that embeddings are bounded and denote

$$M := \sqrt{2} \max_{j \in [m]} \|\bar{\boldsymbol{h}}_j\| \, .$$

Given $\mathcal{T}_m$, let

$$\mathcal{F} = \text{span}\left(\left\{(\boldsymbol{e}_z - \boldsymbol{e}_{z'})\bar{\boldsymbol{h}}_j^{\top} : z \neq z' \in \mathcal{S}_j, j \in [m]\right\}\right)$$

a subspace of $V \times d$ matrices and $\mathcal{F}^{\perp}$ its orthogonal complement. Denote $\mathcal{P}_{\mathcal{F}}, \mathcal{P}_{\perp}$ the orthogonal projections onto $\mathcal{F}$ and $\mathcal{F}^{\perp}$, respectively. For convenience, for $\boldsymbol{W} \in \mathbb{R}^{V \times d}$, we denote

$$\boldsymbol{W}_{\mathcal{F}} := \mathcal{P}_{\mathcal{F}}(\boldsymbol{W}) \qquad \text{and} \qquad \boldsymbol{W}_{\perp} = \mathcal{P}_{\perp}(\boldsymbol{W}) \, .$$

Define

$$\mathrm{CE}_{\mathcal{F}}(\boldsymbol{W}) = \sum_{j \in [m]} \hat{\pi}_j \sum_{z \in \mathcal{S}_j} \hat{p}_{j,z} \log\left(1 + \sum_{z \neq z} e^{-(\boldsymbol{e}_z - \boldsymbol{e}_{z'})^{\top} \boldsymbol{W} \bar{\boldsymbol{h}}_j}\right) \, . \tag{12}$$

Clearly, for all $\boldsymbol{W} \in \mathbb{R}^{V \times d}$, it holds $\mathrm{CE}(\boldsymbol{W}) \geq \mathrm{CE}_{\mathcal{F}}(\boldsymbol{W})$. Note also that for all $\boldsymbol{W} \in \mathcal{F}$ and for all $\boldsymbol{W}^{\mathrm{d}} \in \mathcal{F}^{\perp}$ that satisfy Eq. (6a), it holds $\mathrm{CE}_{\mathcal{F}}(\boldsymbol{W}) = \lim_{R \to \infty} \mathrm{CE}(\boldsymbol{W} + R\boldsymbol{W}^{\mathrm{d}})$. Thus, under NTP compatibility and NTP separability,

$$\inf_{\boldsymbol{W} \in \mathcal{F}} \mathrm{CE}_{\mathcal{F}}(\boldsymbol{W}) = \inf_{\boldsymbol{W}} \mathrm{CE}(\boldsymbol{W}) = \mathcal{H}. \tag{13}$$

# E   Proofs

## E.1   Gradient Descent

Throughout we assume GD is ran with step-size $\eta \leq 1/(2L)$ where $L$ is the smoothness of CE loss. This condition is not explicitly mentioned thereafter.

### E.1.1   Auxiliary Lemmata

The following result follows from standard optimization analysis for smooth convex functions specialized to functions that do not attain their infimum. The version presented here is adopted from Lemma 2 in [37].

**Lemma 3.** *It holds*

$$\lim_{k \to \infty} \mathrm{CE}(\boldsymbol{W}_k) = \inf_{\boldsymbol{W}} \mathrm{CE}(\boldsymbol{W})$$

*and also* $\lim_{k \to \infty} \|\boldsymbol{W}_k\| = \infty$.

In the lemma below, we collect some useful and simple-to-show properties of the GD and regularization paths. These are adaptations of corresponding results for one-hot binary classification over general non-separable data established in [34].

**Lemma 4.** *Suppose conditions* (6) *hold for some* $\boldsymbol{W}^{\mathrm{d}}$. *Also, that there exists* $\boldsymbol{W}^{\mathrm{P}} = \boldsymbol{W}^{\star} \in \mathcal{F}$ *satisfying condition* (4). *The following hold:*

1. $\mathrm{CE}_{\mathcal{F}}(\boldsymbol{W}^{\star}) = \inf_{\boldsymbol{W} \in \mathcal{F}} \mathrm{CE}_{\mathcal{F}}(\boldsymbol{W}) = \mathcal{H}$,

2. $\boldsymbol{W}^{\star}$ *is the* unique *minimizer of* $\mathrm{CE}_{\mathcal{F}}$ *on the subspace $\mathcal{F}$,*

3. $\lim_{k \to \infty} \mathcal{P}_{\mathcal{F}}(\boldsymbol{W}_k) = \boldsymbol{W}^{\star}$, *where $\boldsymbol{W}_k$ are GD iterates,*

4. $\lim_{k \to \infty} \|\mathcal{P}_{\perp}(\boldsymbol{W}_k)\| = \infty$,

5. $\lim_{B \to \infty} \mathcal{P}_{\mathcal{F}}(\widehat{\boldsymbol{W}}_B) = \boldsymbol{W}^{\star}$, *where $\widehat{\boldsymbol{W}}_B$ is the reguarlized solution* (8)*,*

6. $\lim_{B \to \infty} \|\mathcal{P}_{\perp}(\widehat{\boldsymbol{W}}_B)\| = \infty$.

*Proof.* It is easy to check by direct substitution of $\boldsymbol{W}^{\star}$ in (12) and use of (4) that $\mathrm{CE}_{\mathcal{F}}(\boldsymbol{W}^{\star}) = \mathcal{H}$. This and (13) show the first claim.

The first claim shows $\boldsymbol{W}^{\star}$ is a minimizer. Suppose for the sake of contradiction there is a different minimizer $\boldsymbol{W}^{\star} \neq \boldsymbol{W}_1 \in \mathcal{F}$. Then, since $\mathrm{CE}_{\mathcal{F}}(\boldsymbol{W}_1) = \mathcal{H}$, it also holds for $\boldsymbol{W}_R := \boldsymbol{W}_1 + R\boldsymbol{W}^{\mathrm{d}}$ that $\lim_{R \to \infty} \mathrm{CE}(\boldsymbol{W}_R) = \mathcal{H}$. In turn, this implies for all $j \in [m]$:

$$\lim_{R \to \infty} \mathbb{S}_z(\boldsymbol{W}_R \bar{\boldsymbol{h}}_j) = \hat{p}_{j,z}, \forall z \in \mathcal{S}_j, \qquad \text{and} \qquad \lim_{R \to \infty} \mathbb{S}_v(\boldsymbol{W}_R \bar{\boldsymbol{h}}_j) = 0, \forall v \notin \mathcal{S}_j.$$

The first condition gives then that $\boldsymbol{W}_1$ must satisfy (4). Since $\boldsymbol{W}^{\star}$ also satisfies these equations, denoting $\boldsymbol{W}_{\Delta} = \boldsymbol{W}^{\star} - \boldsymbol{W}_1 \neq 0$, it holds:

$$\langle \boldsymbol{W}_{\Delta}, (\boldsymbol{e}_z - \boldsymbol{e}_{z'})^{\top} \bar{\boldsymbol{h}}_j) \rangle = 0, \ \forall j \in [m], z \neq z' \in \mathcal{S}_j.$$

But $\boldsymbol{W}_{\Delta} \in \mathcal{F}$, so this forms a contradiction. Hence, $\boldsymbol{W}^{\star}$ is unique solution in $\mathcal{F}$ of (4) and unique minimizer of $\mathrm{CE}_{\mathcal{F}}$ on the subspace $\mathcal{F}$.

The proof of the third claim follows the same way as the proof of part (1) of Thm. 15 of [37]. For completeness: It follows by the lemma's assumptions and Lemma 3 that $\lim_{k \to \infty} \mathrm{CE}(\boldsymbol{W}_k) = \mathcal{H}$. Combining with the first claim of the lemma yields $\lim_{k \to \infty} \mathrm{CE}(\boldsymbol{W}_k) = \mathrm{CE}_{\mathcal{F}}(\boldsymbol{W}^{\star})$. Since $\mathrm{CE}_{\mathcal{F}}(\boldsymbol{W}_k) \leq \mathrm{CE}(\boldsymbol{W}_k)$, this finally gives

$$\lim_{k \to \infty} \mathrm{CE}_{\mathcal{F}}(\boldsymbol{W}_k) = \lim_{k \to \infty} \mathrm{CE}_{\mathcal{F}}(\mathcal{P}_{\mathcal{F}}(\boldsymbol{W}_k)) = \mathrm{CE}_{\mathcal{F}}(\boldsymbol{W}^{\star}).$$

Since $\boldsymbol{W}^{\star}$ is unique by the second claim, the desired then follows.

For the fourth claim, recall from Lemma 3 that $\lim_{k \to \infty} \|\boldsymbol{W}_k\| = \infty$. From the previous claim, we also have $\lim_{k \to \infty} \|\mathcal{P}_{\mathcal{F}}(\boldsymbol{W}_k)\| < C$ for some constant $C > \|\boldsymbol{W}^{\star}\|$. Thus, the desired follows by applying the fact that $\|\boldsymbol{W}_k\| = \|\mathcal{P}_{\mathcal{F}}(\boldsymbol{W}_k)\| + \|\mathcal{P}_{\perp}(\boldsymbol{W}_k)\|$.

The proof of the last two claim is exactly same as that of the third and fourth claim. Only now use the facts that $\lim_{B \to \infty} \mathrm{CE}(\boldsymbol{W}_B) = \mathcal{H}$ and $\lim_{B \to \infty} \|\boldsymbol{W}_B\| = \infty$ (see proof of Theorem 1). $\qquad \square$

### E.1.2 Key Lemma

**Lemma 5.** *Let $\boldsymbol{W}_k$ denote the GD iterate at iteration $k$. Recall the decomposition $\boldsymbol{W}_k = \mathcal{P}_{\mathcal{F}}(\boldsymbol{W}_k) + \mathcal{P}_{\perp}(\boldsymbol{W}_k) = \boldsymbol{W}_{k,\mathcal{F}} + \boldsymbol{W}_{k,\perp}$. Fix any $\alpha \in (0,1)$. There exists large enough $R = R(\alpha)$ and $k_0 = k_0(R)$ such that for any $k \geq k_0$, it holds that $\|\boldsymbol{W}_{k,\perp}\| \geq R$ and*

$$\mathrm{CE}\left(\boldsymbol{W}_{k,\mathcal{F}} + (1+\alpha)\|\boldsymbol{W}_{k,\perp}\|\overline{\boldsymbol{W}^{\mathrm{mm}}}\right) \leq \mathrm{CE}(\boldsymbol{W}_k). \tag{14}$$

*Proof.* We drop the subscript $k$ to lighten notation.

First, note by Lemma 4.D that, for arbitrary $R$, we can pick $k_1 = k_1(R)$ such that for all $k \geq k_1$: $\|\boldsymbol{W}_{\perp}\| \geq R$.

Thus next, we will prove the main claim, i.e. for large enough $\|\boldsymbol{W}_\perp\|$ inequality (14) holds. Denote $R' = \frac{\|\boldsymbol{W}_\perp\|}{\|\boldsymbol{W}^{\mathrm{mm}}\|}$. Substituting in CE expression (2), and using the fact that $\boldsymbol{W}^{\mathrm{mm}} \in \mathcal{F}^\perp$ by (6a) yield:

$$\mathrm{CE}\big(\boldsymbol{W}_{\mathcal{F}} + (1+\alpha)R'\boldsymbol{W}^{\mathrm{mm}}\big)$$

$$= \sum_{j \in [m]} \hat{\pi}_j \sum_{z \in \mathcal{S}_j} \hat{p}_{j,z} \log\left(\sum_{z' \in \mathcal{S}_j} e^{-(\boldsymbol{e}_z - \boldsymbol{e}_{z'})^\top \boldsymbol{W}_{\mathcal{F}} \bar{\boldsymbol{h}}_j} + \sum_{v \notin \mathcal{S}_j} e^{-(\boldsymbol{e}_z - \boldsymbol{e}_v)^\top \boldsymbol{W}_{\mathcal{F}} \bar{\boldsymbol{h}}_j} + \sum_{v \notin \mathcal{S}_j} e^{-(1+\alpha)R'(\boldsymbol{e}_z - \boldsymbol{e}_v)^\top \boldsymbol{W}^{\mathrm{mm}} \bar{\boldsymbol{h}}_j}\right).$$

$$= \sum_{j \in [m]} \hat{\pi}_j \sum_{z \in \mathcal{S}_j} \hat{p}_{j,z} \log\left(\sum_{v \in \mathcal{V}} e^{-(\boldsymbol{e}_z - \boldsymbol{e}_v)^\top \boldsymbol{W}_{\mathcal{F}} \bar{\boldsymbol{h}}_j} + \sum_{v \notin \mathcal{S}_j} e^{-(1+\alpha)R'(\boldsymbol{e}_z - \boldsymbol{e}_v)^\top \boldsymbol{W}^{\mathrm{mm}} \bar{\boldsymbol{h}}_j}\right). \tag{15}$$

Moreover, decomposing $\boldsymbol{W} = \boldsymbol{W}_{\mathcal{F}} + \boldsymbol{W}_\perp$, and defining

$$\widetilde{\boldsymbol{W}}_\perp := \frac{\|\boldsymbol{W}^{\mathrm{mm}}\|}{\|\boldsymbol{W}_\perp\|}\boldsymbol{W}_\perp = \frac{1}{R}\boldsymbol{W}_\perp,$$

we have

$$\mathrm{CE}\big(\boldsymbol{W}\big) = \sum_{j \in [m]} \hat{\pi}_j \sum_{z \in \mathcal{S}_j} \hat{p}_{j,z} \log\left(\sum_{z' \in \mathcal{S}_j} e^{-(\boldsymbol{e}_z - \boldsymbol{e}_{z'})^\top \boldsymbol{W}_{\mathcal{F}} \bar{\boldsymbol{h}}_j} + \sum_{v \notin \mathcal{S}_j} e^{-(\boldsymbol{e}_z - \boldsymbol{e}_v)^\top \boldsymbol{W}_{\mathcal{F}} \bar{\boldsymbol{h}}_j} + \sum_{v \notin \mathcal{S}_j} e^{-R'(\boldsymbol{e}_z - \boldsymbol{e}_v)^\top \widetilde{\boldsymbol{W}}_\perp \bar{\boldsymbol{h}}_j}\right)$$

$$= \sum_{j \in [m]} \hat{\pi}_j \sum_{z \in \mathcal{S}_j} \hat{p}_{j,z} \log\left(\sum_{v \in \mathcal{V}} e^{-(\boldsymbol{e}_z - \boldsymbol{e}_v)^\top \boldsymbol{W}_{\mathcal{F}} \bar{\boldsymbol{h}}_j} + \sum_{v \notin \mathcal{S}_j} e^{-R'(\boldsymbol{e}_z - \boldsymbol{e}_v)^\top \widetilde{\boldsymbol{W}}_\perp \bar{\boldsymbol{h}}_j}\right), \tag{16}$$

where we used that, by definition, $\boldsymbol{W}_\perp \in \mathcal{F}^\perp$. Thus, our goal becomes showing (15) $\le$ (16), for large enough $R$. To do this, we consider two cases as follows below.

For the remaining of the proof recall $M := \max_{j \in [m]} \sqrt{2}\|\bar{\boldsymbol{h}}_j\|$ and use the logits shorthand:

$$\widetilde{\ell}_{j,v} = \boldsymbol{e}_v^\top \widetilde{\boldsymbol{W}}_\perp \bar{\boldsymbol{h}}_j \qquad \text{and} \qquad \ell_{j,v}^{\mathrm{mm}} = \boldsymbol{e}_v^\top \boldsymbol{W}^{\mathrm{mm}} \bar{\boldsymbol{h}}_j.$$

*Case 1: $\boldsymbol{W}_\perp$ is well aligned with $\boldsymbol{W}^{\mathrm{mm}}$.* Suppose

$$\|\boldsymbol{W}^{\mathrm{mm}} - \widetilde{\boldsymbol{W}}_\perp\| \le \epsilon := \frac{\alpha}{M}. \tag{17}$$

Using this, linearity of logits, and Cauchy-Schwartz, yields

$$\widetilde{\ell}_{j,z} - \widetilde{\ell}_{j,v} \le \ell_{j,z}^{\mathrm{mm}} - \ell_{j,v}^{\mathrm{mm}} + \epsilon M, \quad \forall j \in [m], z \in \mathcal{S}_j, v \notin \mathcal{S}_j.$$

Thus,

$$\sum_{v \notin \mathcal{S}_j} e^{-R'(\boldsymbol{e}_z - \boldsymbol{e}_v)^\top \widetilde{\boldsymbol{W}}_\perp \bar{\boldsymbol{h}}_j} \ge e^{-\epsilon M R'} \sum_{v \notin \mathcal{S}_j} e^{-R'(\boldsymbol{e}_z - \boldsymbol{e}_v)^\top \boldsymbol{W}^{\mathrm{mm}} \bar{\boldsymbol{h}}_j} = e^{-\alpha R'} \sum_{v \notin \mathcal{S}_j} e^{-R'(\boldsymbol{e}_z - \boldsymbol{e}_v)^\top \boldsymbol{W}^{\mathrm{mm}} \bar{\boldsymbol{h}}_j}$$

Also recall by feasibility of $\boldsymbol{W}^{\mathrm{mm}}$ that

$$\ell_{j,z}^{\mathrm{mm}} - \ell_{j,v}^{\mathrm{mm}} \ge 1, \forall j \in [m], z \in \mathcal{S}_j, v \notin \mathcal{S}_j. \tag{18}$$

Thus,

$$\sum_{v \notin \mathcal{S}_j} e^{-(1+\alpha)R'(\boldsymbol{e}_z - \boldsymbol{e}_v)^\top \widetilde{\boldsymbol{W}}_\perp \bar{\boldsymbol{h}}_j} \le e^{-\alpha R'} \sum_{v \notin \mathcal{S}_j} e^{-R'(\boldsymbol{e}_z - \boldsymbol{e}_v)^\top \boldsymbol{W}^{\mathrm{mm}} \bar{\boldsymbol{h}}_j}$$

Comparing the above two displays yields

$$\sum_{v \notin \mathcal{S}_j} e^{-(1+\alpha)R'(\boldsymbol{e}_z - \boldsymbol{e}_v)^\top \widetilde{\boldsymbol{W}}_\perp \bar{\boldsymbol{h}}_j} \le \sum_{v \notin \mathcal{S}_j} e^{-R'(\boldsymbol{e}_z - \boldsymbol{e}_v)^\top \widetilde{\boldsymbol{W}}_\perp \bar{\boldsymbol{h}}_j},$$

which implies the desired (15)$\le$(16) for any value of $R'$ (eqv. $\|\boldsymbol{W}_\perp\|$).

*Case 2: No alignment.* Suppose now that (17) does not hold. Note that $\|\widetilde{\boldsymbol{W}}_\perp\| = \|\boldsymbol{W}^{\mathrm{mm}}\|$ and since (NTP-SVM) has a unique solution it must be that $\widetilde{\boldsymbol{W}}_\perp$ is not feasible. But $\widetilde{\boldsymbol{W}}_\perp \in \mathcal{F}_\perp$, thus it satisfies the equality constraints. This then means that there exist $\delta := \delta(\epsilon)$ and $j_\star \in [m], v_\star \notin \mathcal{S}_{j_\star}$ such that

$$\widetilde{\ell}_{j_\star, z} - \widetilde{\ell}_{j_\star, v_\star} \le 1 - \delta, \quad \forall z \in \mathcal{S}_{j_\star}. \tag{19}$$

(Note the above holds for all $z \in \mathcal{S}_{j_\star}$ because $\widetilde{\ell}_{j_\star,z} = \widetilde{\ell}_{j_\star,z'}$ since $\widetilde{\boldsymbol{W}}_\perp \in \mathcal{F}_\perp$.)

To continue, we introduce the shorthand notation

$$A_{j,z} := A_{j,z}(\boldsymbol{W}) = \sum_{v \in \mathcal{V}} e^{-(\boldsymbol{e}_z - \boldsymbol{e}_v)^\top \boldsymbol{W}_\mathcal{F} \bar{\boldsymbol{h}}_j}$$

as well as

$$A_{\min} := \min_{j \in [m], z \in \mathcal{S}_j} A_{j,z}, \qquad \text{and} \qquad A_{\max} := \max_{j \in [m], z \in \mathcal{S}_j} A_{j,z}.$$

Using (19) we may lower bound (16) as follows:

$$\mathrm{CE}(\boldsymbol{W}) - \sum_{j \in [m]} \hat{\pi}_j \sum_{z \in \mathcal{S}_j} \hat{p}_{j,z} \log \left( \sum_{v \in \mathcal{V}} e^{-(\boldsymbol{e}_z - \boldsymbol{e}_v)^\top \boldsymbol{W}_\mathcal{F} \bar{\boldsymbol{h}}_j} \right) \geq \hat{\pi}_{j_\star} \sum_{z \in \mathcal{S}_j} \hat{p}_{j,z} \log \left( 1 + \frac{e^{-R'(\boldsymbol{e}_z - \boldsymbol{e}_{v_\star})^\top \widetilde{\boldsymbol{W}}_\perp \bar{\boldsymbol{h}}_{j_\star}}}{A_{j_\star,z}} \right)$$

$$\geq \hat{\pi}_{j_\star} \sum_{z \in \mathcal{S}_j} \hat{p}_{j,z} \log \left( 1 + \frac{e^{-R'(1-\delta)}}{A_{\max}} \right)$$

$$\geq \frac{e^{-R'(1-\delta)}}{n(A_{\max} + 1)}, \tag{20}$$

where in the last line we used $\hat{\pi}_j \geq 1/n, \forall j \in [m]$ as well as $\log(1 + x) \geq \frac{x}{1+x}, x > 0$.

On the other hand, using property (18) for max-margin logits, we can upper bound (15) as follows:

$$\mathrm{CE}\left( \boldsymbol{W}_\mathcal{F} + (1 + \alpha) R' \boldsymbol{W}^{\mathrm{mm}} \right) - \sum_{j \in [m]} \hat{\pi}_j \sum_{z \in \mathcal{S}_j} \hat{p}_{j,z} \log \left( \sum_{v \in \mathcal{V}} e^{-(\boldsymbol{e}_z - \boldsymbol{e}_v)^\top \boldsymbol{W}_\mathcal{F} \bar{\boldsymbol{h}}_j} \right) \leq \log \left( 1 + \frac{V e^{-R'(1+\alpha)}}{A_{\min}} \right)$$

$$\leq \frac{V e^{-R'(1+\alpha)}}{A_{\min}}, \tag{21}$$

where in the last line we used $\log(1 + x) \leq x, x > 0$.

In view of the two last displays, it suffices that

$$V \frac{e^{-R'(1+\alpha)}}{A_{\min}} \leq \frac{e^{-R'(1-\delta)}}{n(A_{\max} + 1)} \iff R' \geq \frac{1}{\delta + \alpha} \log \left( \frac{nV(A_{\max} + 1)}{A_{\min}} \right).$$

All it remains is obtaining bounds for $A_{\min}, A_{\max}$ specifically showing that they do not depend on $R$. By Cauchy-Schwartz:

$$V e^{-M \|\boldsymbol{W}_\mathcal{F}\|} \leq \boldsymbol{A}_{\min} \leq \boldsymbol{A}_{\max} \leq V e^{M \|\boldsymbol{W}_\mathcal{F}\|}$$

Further recall by Lemma 4.C that if $k$ is large enough then

$$\|\boldsymbol{W}_\mathcal{F} - \boldsymbol{W}^\star\| \leq \|\boldsymbol{W}^\star\| \implies \|\boldsymbol{W}_\mathcal{F}\| \leq 2\|\boldsymbol{W}^\star\|. \tag{22}$$

Thus, there exists $k_\star = k_\star(\|\boldsymbol{W}_\star\|)$ such that for all $k \geq k_\star$:

$$V e^{-2M \|\boldsymbol{W}_\star\|} \leq \boldsymbol{A}_{\min} \leq \boldsymbol{A}_{\max} \leq V e^{2M \|\boldsymbol{W}_\star\|}.$$

Hence, the desired (21)≤(20) holds provided

$$\|\boldsymbol{W}_\perp\| \geq \frac{\|\boldsymbol{W}^{\mathrm{mm}}\|}{\alpha} \log \left( 2nV e^{4\|\boldsymbol{W}^\star\|} \right). \tag{23}$$

Set $R = R(\alpha) = \{\text{RHS of (23)}\}$ and $k_0(R) := \max\{k_1(R), k_\star\}$. We have shown this guarantees for all $k \geq k_0$: $\|\boldsymbol{W}_\perp\| \geq R$ and by choice of $R$ also (21)≤(20). This in turn implies (15)≤(16), as desired to complete the proof. $\qquad\square$

### E.1.3 Proof of Theorem 2

For the subspace component, see Lemma 4.C. For the directional convergence, the key ingredient of the proof is Lemma 5. After that, the proof follows identically to Thm. 15(2) in [37]. We include the details for completeness, but there are no novel aspects in the rest of this section.

Let any $\epsilon \in (0,1)$ and choose $\alpha = \epsilon/(1-\epsilon)$. By Lemma 5, there exists $k_0$ such that for any $k \geq k_0$, we have

$$\|\boldsymbol{W}_{k,\perp}\| \geq \max\{R(\alpha), 1/2\}$$

and

$$
\begin{aligned}
\langle \nabla \operatorname{CE}(\boldsymbol{W}_k), \boldsymbol{W}_{k,\perp} - (1+\alpha)\|\boldsymbol{W}_{k,\perp}\|\overline{\boldsymbol{W}^{\mathrm{mm}}}\rangle &= \langle \nabla \operatorname{CE}(\boldsymbol{W}_k), \boldsymbol{W}_k - (\boldsymbol{W}_{k,\mathcal{F}} + (1+\alpha)\|\boldsymbol{W}_{k,\perp}\|\overline{\boldsymbol{W}^{\mathrm{mm}}})\rangle \\
&\geq \operatorname{CE}(\boldsymbol{W}_k) - \operatorname{CE}(\boldsymbol{W}_{k,\mathcal{F}} + (1+\alpha)\|\boldsymbol{W}_{k,\perp}\|\overline{\boldsymbol{W}^{\mathrm{mm}}}) \geq 0\,,
\end{aligned}
$$

where we also used convexity of the loss.

Consequently,

$$
\begin{aligned}
\langle \boldsymbol{W}_{k+1} - \boldsymbol{W}_k, \overline{\boldsymbol{W}^{\mathrm{mm}}}\rangle &= \langle -\eta\nabla \operatorname{CE}(\boldsymbol{W}_k), \overline{\boldsymbol{W}^{\mathrm{mm}}}\rangle \\
&\geq (1-\epsilon)\langle -\eta\nabla \operatorname{CE}(\boldsymbol{W}_k), \overline{\boldsymbol{W}_{k,\perp}}\rangle \\
&\geq (1-\epsilon)\langle \boldsymbol{W}_{k+1,\perp} - \boldsymbol{W}_{k,\perp}, \overline{\boldsymbol{W}_{k,\perp}}\rangle \\
&\geq (1-\epsilon)\langle \boldsymbol{W}_{k+1,\perp} - \boldsymbol{W}_{k,\perp}, \overline{\boldsymbol{W}_{k,\perp}}\rangle \\
&= \frac{(1-\epsilon)}{2\|\boldsymbol{W}_{k,\perp}\|}\left(\|\boldsymbol{W}_{k+1,\perp}\|^2 - \|\boldsymbol{W}_{k,\perp}\|^2 - \|\boldsymbol{W}_{k+1,\perp} - \boldsymbol{W}_{k,\perp}\|^2\right) \\
&\geq (1-\epsilon)\left(\|\boldsymbol{W}_{k+1,\perp}\| - \|\boldsymbol{W}_{k,\perp}\| - 2\eta(\operatorname{CE}(\boldsymbol{W}_{k,\perp}) - \operatorname{CE}(\boldsymbol{W}_{k+1,\perp}))\right),
\end{aligned}
$$

where the last step used $\|\boldsymbol{W}_{k,\perp}\| \geq 1/2$, the fact that $x^2 - y^2 \geq 2y(x-y), \forall x,y$ and smoothness of the CE loss.

Telescoping the above expression and rearranging yields

$$
\begin{aligned}
\langle \overline{\boldsymbol{W}}_k, \overline{\boldsymbol{W}^{\mathrm{mm}}}\rangle &\geq (1-\epsilon)\frac{\|\boldsymbol{W}_{k,\perp}\|}{\|\boldsymbol{W}_k\|} - \frac{\langle \boldsymbol{W}_{k_0}, \overline{\boldsymbol{W}^{\mathrm{mm}}}\rangle - (1-\epsilon)\|\boldsymbol{w}_{k_0,\perp}\| - \eta\operatorname{CE}(\boldsymbol{W}_{k_0})}{\|\boldsymbol{W}_k\|} \\
&\geq (1-\epsilon) - \frac{\|\boldsymbol{W}_{k,\mathcal{F}}\|_2 + \langle \boldsymbol{W}_{k_0}, \overline{\boldsymbol{W}^{\mathrm{mm}}}\rangle - (1-\epsilon)\|\boldsymbol{w}_{k_0,\perp}\| - \eta\operatorname{CE}(\boldsymbol{W}_{k_0})}{\|\boldsymbol{W}_k\|}
\end{aligned}
$$

Now recall from Lemma 4 that $\lim_{k\to\infty}\|\boldsymbol{W}_k\| = \infty$ and $\lim_{k\to\infty}\|\boldsymbol{W}_{k,\mathcal{F}}\| = \|\boldsymbol{W}^\star\|$. Thus, $\liminf_{k\to\infty}\langle \overline{\boldsymbol{W}}_k, \overline{\boldsymbol{W}^{\mathrm{mm}}}\rangle \geq 1 - \epsilon$. Since $\epsilon$ is arbitrary, the desired follows.

## E.2 Regularization Path

We provide a detailed proof of Theorem 1 filling in missing details from the proof sketch in the main paper.

### E.2.1 Proof of Theorem 1

First, we show that $\widehat{\boldsymbol{W}}_B$ is on the boundary, i.e. $\|\widehat{\boldsymbol{W}}_B\| = B$. Suppose not, then $\langle \nabla \operatorname{CE}(\widehat{\boldsymbol{W}}_B), \boldsymbol{U}\rangle = 0$ for all $\boldsymbol{U} \in \mathbb{R}^{V\times d}$. Using the CE expression in (2) and a few algebraic manipulations, yields

$$\langle -\nabla \operatorname{CE}(\widehat{\boldsymbol{W}}_B), \boldsymbol{U}\rangle = \sum_{j\in[m]} \hat{\pi}_j \sum_{z\in\mathcal{S}_j} \hat{p}_{j,z}\Big(\sum_{\substack{z'\in\mathcal{S}_j \\ z'\neq z}} s_{j,z'}(\boldsymbol{e}_z - \boldsymbol{e}_{z'})^\top \boldsymbol{U}\bar{\boldsymbol{h}}_j + \sum_{v\notin\mathcal{S}_j} s_{j,v}(\boldsymbol{e}_z - \boldsymbol{e}_v)^\top \boldsymbol{U}\bar{\boldsymbol{h}}_j\Big), \quad (24)$$

where we denote the output probabilities at $\widehat{\boldsymbol{W}}_B$ as $s_{j,v} := \mathbb{S}_v(\widehat{\boldsymbol{W}}_B\bar{\boldsymbol{h}}_j), v \in \mathcal{V}, j \in [m]$. Choose $\boldsymbol{U} = \boldsymbol{W}^{\mathrm{mm}}$ in (24). Then, the first term in the parenthesis in (24) is zero by (6a), while the second term is strictly positive by (6b) and strict positivity of softmax entries, leading to contradiction.

Now, consider point $\boldsymbol{W}_B^\star = \boldsymbol{W}^\star + R(B)\cdot\boldsymbol{W}^{\mathrm{mm}}$, where, $\boldsymbol{W}^\star \in \mathcal{F}$ satisfies (4), and $R = R(B)$ is chosen such that $\|\boldsymbol{W}_B^\star\| = B$. Concretely, for $B > \|\boldsymbol{W}^\star\|$, set

$$R = \frac{1}{\|\boldsymbol{W}^{\mathrm{mm}}\|}\sqrt{B^2 - \|\boldsymbol{W}^\star\|^2}.$$

Note also that $R/B \to 1/\|\boldsymbol{W}^{\mathrm{mm}}\|$ as $B \to \infty$. We will show that $\boldsymbol{W}_B^\star$ attains a small CE loss as $B$ (hence, $R$) grows. To do this, denote for convenience the logits for all $v \in \mathcal{V}, j \in [m]$:

$$\ell_{j,v}^\star := \boldsymbol{e}_v^\top \boldsymbol{W}^\star \bar{\boldsymbol{h}}_j \quad \text{and} \quad \ell_{j,v}^{\mathrm{mm}} := \boldsymbol{e}_v^\top \boldsymbol{W}^{\mathrm{mm}}\bar{\boldsymbol{h}}_j\,,$$

and note that $e_v^\top W_B^\star \bar{h}_j = \ell_{j,v}^\star + R\ell_{j,v}^{\mathrm{mm}}$. By using (4) and (6a):

$$\sum_{z' \in \mathcal{S}_j} e^{-(\ell_{j,z}^\star + R\ell_{j,z}^{\mathrm{mm}} - \ell_{j,z'}^\star - R\ell_{j,z'}^{\mathrm{mm}})} = \frac{1}{\hat{p}_j}.$$

Moreover, using (6b)

$$\sum_{v \notin \mathcal{S}_j} e^{-(\ell_{j,z}^\star + R\ell_{j,z}^{\mathrm{mm}} - \ell_{j,v}^\star - R\ell_{j,v}^{\mathrm{mm}})} \le e^{-R} \sum_{v \notin \mathcal{S}_j} e^{-(\ell_{j,z}^\star - \ell_{j,v}^\star)} \le C\, e^{-R},$$

where we define constant (independent of $R$) $C := V e^{\|W^\star\| M}$, for $M := \sqrt{2} \cdot \max_{j \in [m]} \|\bar{h}_j\|$.

Combining the above displays and using in Eq. (2), yields

$$\mathrm{CE}(W_B^\star) \le \sum_{j \in [m]} \hat{\pi}_j \sum_{z \in \mathcal{S}_j} \hat{p}_{j,z} \log \Big(\frac{1}{\hat{p}_{j,z}} + C\, e^{-R}\Big) \le \sum_{j \in [m]} \hat{\pi}_j \sum_{z \in \mathcal{S}_j} \hat{p}_{j,z} \Big(\log \Big(\frac{1}{\hat{p}_{j,z}}\Big) + \hat{p}_{j,z} C\, e^{-R}\Big)$$

$$\le \mathcal{H} + C\, e^{-R}, \tag{25}$$

where, the second line uses $\log(1 + x) \le x, x > 0$, and the third line uses $\hat{\pi}_j, \hat{p}_{j,z}$ are probabilities.

Next, towards arriving at a contradiction, we will show that if $\widehat{W}_B$ is not in the direction of $W^{\mathrm{mm}}$, then it incurs a loss that is larger than $\mathrm{CE}(W_B^\star)$. Concretely, assuming the statement of the theorem is not true, we we will upper bound

$$\mathrm{CE}(\widehat{W}_B) - \mathcal{H} = \sum_{j \in [m]} \hat{\pi}_j \sum_{z \in \mathcal{S}_j} \hat{p}_{j,z} \log \Big(\frac{\hat{p}_{j,z}}{\mathbb{S}_z(\widehat{W}_B \bar{h}_j)}\Big). \tag{26}$$

By our assumption, there exists $\epsilon > 0$, such that there exists arbitrarily large $B$ satisfying:

$$\Big\|\frac{\|W^{\mathrm{mm}}\|}{B} \widehat{W}_B - W^{\mathrm{mm}}\Big\| > \epsilon. \tag{27}$$

Define

$$\widehat{W} = \frac{1}{R'(B)} \big(\widehat{W}_B - W^\star\big),$$

where, $R' = R'(B) > 0$ is chosen so that $\|\widehat{W}\| = \|W^{\mathrm{mm}}\|$. Concretely, for large enough $B \ge 2\|W^\star\|$, set

$$R' = \frac{1}{\|W^{\mathrm{mm}}\|} \sqrt{B^2 - 2B\langle \overline{W}_B, W^\star\rangle + \|W^\star\|^2}\,.$$

Note that it holds $\lim_{B \to \infty} R'/B = 1/\|W^{\mathrm{mm}}\|$. Thus, we can always choose $B$ large enough so that Eq. (27) guarantees $\|\widehat{W} - W^{\mathrm{mm}}\| \ge \epsilon'$, for some $\epsilon' > 0$. Since $W^{\mathrm{mm}}$ is the unique minimizer of (NTP-SVM) and $\|\widehat{W}\| = \|W^{\mathrm{mm}}\|$, it follows that there exists $\delta \in (0, 1)$ and $j \in [m]$ such that at least one of the following is true

(i) $\exists z$ and $z' \ne z \in \mathcal{S}_j$ such that

$$|(e_z - e_{z'})^\top \widehat{W} \bar{h}_j| \ge \delta\,, \tag{28}$$

(ii) $\exists z \in \mathcal{S}_j, v \notin \mathcal{S}_j$ such that

$$(e_z - e_v)^\top \widehat{W} \bar{h}_j \le 1 - \delta. \tag{29}$$

*Case (i):* Without loss of generality $(e_z - e_{z'})^\top \widehat{W} \bar{h}_j \le -\delta$ (otherwise, flip $z, z'$). Thus, ignoring all but one term in (26) gives

$$\mathrm{CE}(\widehat{W}_B) - \mathcal{H} \ge \hat{\pi}_j \hat{p}_{j,z} \log \Big(\frac{\hat{p}_{j,z}}{\mathbb{S}_z(\widehat{W}_B \bar{h}_j)}\Big) \ge \hat{\pi}_j \hat{p}_{j,z} \log \Big(\hat{p}_{j,z} e^{(\ell_{j,z'} - \ell_{j,z})}\Big), \tag{30}$$

where we use $\ell_{j,v} = e_v^\top \widehat{W}_B \bar{h}_j, v \in \mathcal{V}$ to denote logits of $\widehat{W}_B$. Using (4) and (28), yields

$$\ell_{j,z'} - \ell_{j,z} = (e_{z'} - e_z)^\top \big(R' \widehat{W} + W^\star\big) \bar{h}_j \ge R'\delta + \log \Big(\frac{\hat{p}_{j,z'}}{\hat{p}_{j,z}}\Big).$$

Put in (26) and using $\hat{p}_{j,z} \geq \hat{\pi}_j \hat{p}_{j,z} \geq 1/n$ shows

$$\mathrm{CE}(\widehat{\boldsymbol{W}}_B) \geq \mathcal{H} + \frac{1}{n} \log\left(\frac{e^{R'\delta}}{n}\right)$$

Compare this with (25). For large enough $B$, it is clear that $\hat{\pi}_j \hat{p}_{j,z} \log\left(\hat{p}_{j,z}\, c\, e^{R'\delta}\right) > Ce^{-R}$. Thus, $\mathrm{CE}(\widehat{\boldsymbol{W}}_B) > \mathrm{CE}(\boldsymbol{W}_B^\star)$, a contradiction.

*Case (ii):* We can assume $\widehat{\boldsymbol{W}} \in \mathcal{F}_\perp$, since otherwise we are in Case (i). Now, again ignoring all but the $(j, z)$ term in the CE loss for which (29) holds for some $v \notin \mathcal{S}_j$, we find

$$\mathrm{CE}(\widehat{\boldsymbol{W}}_B) - \mathcal{H} \geq \hat{\pi}_j \hat{p}_{j,z} \log\left(\hat{p}_{j,z}\Big(\sum_{z' \in \mathcal{S}_j} e^{(\ell_{j,z'} - \ell_{j,z})} + e^{(\ell_{j,v} - \ell_{j,z})}\Big)\right).$$

Using $\mathcal{P}_\mathcal{T}(\widehat{\boldsymbol{W}}_B) = \boldsymbol{W}^\star$ yields

$$\sum_{z' \in \mathcal{S}_j} e^{(\ell_{j,z'} - \ell_{j,z})} = \sum_{z' \in \mathcal{S}_j} \frac{\hat{p}_{j,z'}}{\hat{p}_{j,z}} = \frac{1}{\hat{p}_{j,z}}\,.$$

Moreover, by (29):

$$e^{\ell_{j,v} - \ell_{j,z}} \geq e^{-R'(1-\delta)} e^{\ell_{j,v}^\star - \ell_{j,z}^\star} \geq c' e^{-R'(1-\delta)},$$

for constant (independent of $B$) $c' := e^{-\|\boldsymbol{W}^\star\|M}$. Putting the above together yield:

$$\mathrm{CE}(\widehat{\boldsymbol{W}}_B) - \mathcal{H} \geq \hat{\pi}_j \hat{p}_{j,z} \log\left(1 + \hat{p}_{j,z} c' e^{-R'(1-\delta)}\right) \geq \frac{c' e^{-R'(1-\delta)}}{2n^2}\,.$$

where the second inequality uses $\log(1 + x) \geq \frac{x}{1+x}, x > 0$.

Compare this with (25). For large enough $B$, (recall $R, R'$ grow at the same rate) it holds $\frac{c'}{2n^2} e^{-R'(1-\delta)} > Ce^{-R}$. Thus, $\mathrm{CE}(\widehat{\boldsymbol{W}}_B) > \mathrm{CE}(\boldsymbol{W}_B^\star)$, a contradiction.

In either case, we arrive at a contradiction, which completes the proof.

