# OpenReview forum: "Implicit Optimization Bias of Next-token Prediction in Linear Models"
_NeurIPS.cc/2024/Conference — NeurIPS 2024 poster_

### Official Review · Reviewer_k4Ko · 2024-06-23

**Soundness:** 3
**Presentation:** 2
**Contribution:** 2
**Rating:** 5
**Confidence:** 3

**Summary:**

This paper studies the implicit bias of the gradient descent on the Next-Token Prediction (LTP) problem in linear models. They first formulate this NTP problem as minimizing the cross-entropy (CE) loss over distinct contexts, each tied with a sparse conditional probability over the token space. They then provide the necessary conditions for the CE loss to reach the entropy lower bound, i.e., the NTP-compatible condition and the NTP-separable condition. Then, they prove one sufficient condition for those two conditions is oevrparameterization, i.e., the dimension of the embedding space d is larger than the number of distinct contexts in the dataset. Assuming both compatible and separable conditions, they then prove the directional convergence of the minimizer of the CE loss within a certain range and the directional convergence of the GD iterate towards the direction of the solution of an NTP-SVM.

In general, I think this paper delves into a good and important problem: the optimization path and implicit bias of NTP mechanism. The authors provided a good formulation, and the proof is solid.

**Strengths:**

1. They investigate an interesting and important problem: the optimization path and the implicit bias of NTP.

2. Their formulation of NTP into the CE minimization over distinct contexts is novel.

3. They provide rigorous theoretical results and the proofs are solid, to my knowledge.

**Weaknesses:**

1. The main issue of this paper is that, for the NTP-compatible and separable conditions to hold, one needs d > m. Does this overparametrization condition usually hold in practice or not? To my knowledge, in practice, the embedding dimension d is much smaller than the number of training data. Since m is not the number of training data and can be much smaller than that, it is not clear to me whether this assumption is possible in practice.

2. There are some paragraphs that are not very clearly written. For example, in lines 154-157, why does equation 4 constrain W^p w.r.t. this subspace? Why is the solution W* unique, assuming equation 4 has a solution? I think those can be expressed as lemmas to make them clearer. In line 148, the authors claim that (3a) holds if and only if the data satisfies the NTP-compatible condition. The 'if' direction is trivial, but the other direction needs a more rigorous proof.

**Questions:**

See above.

**Limitations:**

See above.

---

> ### Author Rebuttal · Authors · 2024-08-05
>
> Thank you for your thoughtful review and the constructive questions raised.  We appreciate the careful read.
> We hope that our responses below answer your questions.
>
> **Q1.  for the NTP-compatible and separable conditions to hold, one needs d > m.**
>
> Here are the key points to consider about the condition $d>m$ of Lemma 1:
>
> 1. First, this is only a sufficient condition, not a necessary one. In general, whether the conditions are satisfied depends on the geometry of embeddings and the sparsity pattern of the language training set.
>
> 2. Second, the result holds for linear models (fixed embeddings). It is an open question (Line 358-360) for future work to derive sufficient conditions for nonlinear models. The NTP-compatibility/separability conditions, which are themselves both necessary and sufficient for the NTP training loss to reach its lower bound, independent of linearity, provide the appropriate formulations for such an endeavor.
>
> Specifically, in nonlinear models, the total number of parameters can be increased by augmenting the width or depth, rather than directly modifying the embedding dimension $d$ as in linear models. This is similar to known results for one-hot classification: For linear models, $d > m$ is a sufficient condition for separability of data (where $m$ is the number of samples and $d$ the ambient dimension), but this condition is relaxed when adding one hidden nonlinear MLP layer, where data can be separable provided $d > m/W$ (with $W$ being the width of the hidden layer). By increasing the width and making $dW$ the total number of parameters, the requirement on $d$ is relaxed. Deriving analogous results for the NTP setting is an interesting future direction. This remains non-trivial since an MLP layer shall be replaced by a sequence-to-sequence architecture (such as transformer), adding complexity to the analysis. Additionally, we expect that the result depends on the sparsity patterns of the support sets.
>
> We appreciate the question and will elaborate on Lines 358-360 in the future work section towards relaxing the current limitation of $d>m$ due to linearity.
>
> **Q2. In lines 154-157, why does equation 4 constrain $W^p$ w.r.t. this subspace? Why is the solution $W^\star$ unique, assuming equation 4 has a solution?**
>
> Assuming Eq. (4) has a solution, say $W_1$, every other solution takes the form
> $W^p = W_1 + W_{\text{null}}$, where
> $W_{\text{null}}$ (the null-space component) is orthogonal to
> $(e_z - e_{z'})h_j^T : z \neq z' \in S_j, j \in [m].$
> Thus,
> $W_{\text{null}} \in F^\perp$. We will reword Line 155 according to this.
>
> Assuming it exists, the solution to Eq. (4) is unique *when constrained to the data subspace $F$*. To see this suppose $W_1$ and
> $W_2$ are both solutions in $F$. Since both are solutions, $W_1 - W_2$ is orthogonal to $F$. However, because $F$ is a subspace, $W_1 - W_2$  must belong to $F$.
> Therefore, $W_1 = W_2$, which is a contradiction. We will include this in the appendix for completeness.
>
>
> **Q3. In line 148, the authors claim that (3a) holds if and only if the data satisfies the NTP-compatible condition. The 'if' direction is trivial, but the other direction needs a more rigorous proof.**
>
> Thank you for highlighting this. Upon review, we agree that our initial statement in line 148 requires more precise phrasing. It is straightforward to see that (3a) implies NTP-compatibility as described in Equation (4). However, the converse is not inherently true; NTP-compatibility alone does not ensure (3a) holds.
>
> For (3a) to be valid, both NTP-separability conditions (Equations 6a and 6b) must also be satisfied in addition to Equation (4). Here’s a proof outline:
>
> Assuming Equations (4), (6a), and (6b) hold for some  $W^p$ and $W^d$, we define $W_\gamma=W^p+\gamma W^d$.
> Decompose the inverse of $S_z(W_\gamma h_j)$ as:
> $$
> 1/S_z(W_\gamma h_j) = 1+ \sum_{z'\neq z\in S_j }\exp\left( (e_z'-e_z)^TW_\gamma h_j \right) + \sum_{v\not\in S_j }\exp\left( (e_v-e_z)^TW_\gamma h_j \right)
> $$
>
> Using the properties from (4) and (6a) for in-support pairs:
> $$
> (e_z'-e_z)^TW_\gamma h_j = \log\left(p_{j,z'}/p_{j,z}\right)
> $$
> leading to:
> $$1+ \sum_{z'\neq z\in S_j }\exp\left( (e_z'-e_z)^TW_\gamma h_j \right)  = 1+\sum_{z'\neq z\in S_j } p_{j,z'}/p_{j,z} = 1+(1-p_{j,z})/p_{j,z}=1/p_{j,z}.$$
>
> For out-of-support pairs, applying (6b) and considering the limit as $\gamma\rightarrow\infty$, we have:
> $$
> \exp\left( (e_v-e_z)^TW_\gamma h_j \right) = \exp\left( \gamma (e_v-e_z)^TW^d h_j \right)\cdot \exp\left( (e_v-e_z)^TW^p h_j \right) \leq \exp\left(-\gamma\right)\cdot \exp\left( (e_v-e_z)^TW^p h_j \right)  \stackrel{\gamma\rightarrow \infty}{\longrightarrow} 0
> $$
>
> Putting these together gives $1/S_z(W_\gamma h_j) \stackrel{\gamma\rightarrow \infty}{\longrightarrow} 1/p_{j,z}$ as desired.
>
> We will include the above calculations within a formal proof of Proposition 1 in the appendix and we will remove the 'iff' statement from Line 148. Thanks again for catching this!

---

> > ### Author Response · Authors · 2024-08-13
> >
> > Dear Reviewer,
> >
> > Thank you again for your thoughtful comments and careful read. Did our responses answer your questions? We look forward to your feedback before the discussion period ends!

---

### Official Review · Reviewer_kgdu · 2024-07-13

**Soundness:** 3
**Presentation:** 3
**Contribution:** 3
**Rating:** 7
**Confidence:** 3

**Summary:**

This work studies the implicit bias of optimization in next token prediction tasks by analyzing the structure of the decoding matrix at infinite time. The paper introduces two novel conditions under which the loss reaches its minimum theoretical value and demonstrates that if these conditions hold (which can be, for example, the case when the model is overparameterized), then after GD training, the decoding matrix will converge (in direction) to a matrix reminiscent of the maximum-margin matrix in "standard" classification.

**Strengths:**

This work studies a timely topic (next token prediction) and approaches it from a learning theoretic perspective (implicit bias of optimization), which has proven to be very fruitful in "standard" classification. The assumption of sparse contexts is clever and should be of wider applicability. The results are novel and analogous to similar results that were proven for "standard" classification. Furthermore, the presentation is comprehensive, with many pointers to related work, which help contextualize this paper's contributions.

**Weaknesses:**

A weakness, which the authors do acknowledge in their work, that prevented me from giving a higher score is that there is no clear connection between the structure of the weights and generalization, as there exists in "standard"/one-hot classification. As a result, it is unclear how much insight can be derived from the current result. I would appreciate the authors' thoughts on this.

Minor: The text is too dense in places, with the authors trying to include more details than what the space permits. I would suggest moving some of the discussion in Sections 6 and 7 to the Appendix to facilitate a smoother flow.

**Questions:**

A minor suggestion: lines 32-34 appear to require rephrasing.

**Limitations:**

The authors thoroughly discuss the limitations of their work.

---

> ### Author Rebuttal · Authors · 2024-08-06
>
> We are grateful for your encouraging feedback and for endorsing our paper.
>
> **Q: A weakness, which the authors do acknowledge in their work, that prevented me from giving a higher score is that there is no clear connection between the structure of the weights and generalization, as there exists in "standard"/one-hot classification. As a result, it is unclear how much insight can be derived from the current result. I would appreciate the authors' thoughts on this.**
>
> Indeed, more work is needed to connect the structure of the weights to generalization. In fact, the question of generalization is at the center of what we envision as follow-up work. Although much of this is still in early stages, we would like to share some of our thinking around linking implicit-bias optimization results to generalization in the language setting below.
>
> The first line of thinking relates to how to quantify generalization. One option is test loss (NTP loss with expectation over fresh contexts). In this case, we have reason to believe that it might be possible to extend techniques from standard classification that relate the generalization gap to training loss via algorithmic stability, e.g. [SK22]. The way the ‘structure of the weights’ would come into such a result is by ensuring that there exist weights, sufficiently close to initialization, such that training loss becomes $\epsilon$-close to its entropy lower bound. Since approximating the lower bound requires NTP-separability, we expect the max-margin weights to give the best candidate for selecting those weights. However, two challenges need to be resolved with this approach: the first is technical and involves extending the above-mentioned prior work to accommodate the imbalanced distribution of (distinct) contexts, which breaks the ‘identically distributed’ assumption. The second is more conceptual and relates to the second line of thinking discussed below: what is a ‘good’ model for the context embeddings and for the sparsity patterns?
> Beyond test loss, it is unclear to us what the right analogue of one-hot error in standard classification is. A possibility is to study specific tasks for which such a metric is clearly defined. A concrete example could be the bracket-tracing task Dyck, for which [Mur+23] recently demonstrated that NTP-trained transformers can generalize well, a phenomenon they call structural grokking, but only after extensive training. We conjecture that the source of inductive bias for this structural grokking phenomenon is actually the NTP training, and that the framing of NTP in our paper might provide the right tools to study this since: (1) the model of NTP as sparse soft-label classification fits the synthetic language dataset Dyck, for which the experiments are reported in [Mur+23]. (2) It is experimentally suggested that structural grokking occurs only after extended training and only after the NTP loss saturates (i.e., when NTP-separability and compatibility conditions kick in). (3) The principle of margin-maximization has been very recently connected to the grokking phenomenon, although only in one-hot settings of modular arithmetic [Moh+23, Mor+24].
>
> The second line of thinking  involves identifying appropriate models for context embeddings and for sparsity patterns of the next-token distribution. Directly modeling context embeddings appears necessary if we are to hope that linear models can still provide (to some extent) insights on language generalization. Simply put the question becomes: what is the simplest (discrete) analogue of the Gaussian mixture model (features distributed normally around a mean vector for each class) which we often use to model image-classification data and produces insights on things like benign-overfitting and optimal loss functions even within linear models? Even if we decide we need to push beyond linear models, we still need to consider modeling the sparsity patterns of the next-token distributions, as this is critical in determining conditions for reaching the entropy lower bound and determining the structure of weights via the NTP max-margin program.
>
> Finally, we hypothesize that considering the conceptual meaning that certain weights might carry could be a way forward. Concretely, decoder weights correspond to word embeddings and last-layer activations ($h_j$ in the paper) correspond to context embeddings. Identifying the structure of these weights translates to the structure of word and context embeddings (see Lines 361-7 for a possible way for future work to arrive at these). Although how to explicitly relate this structure to generalization is at the moment still an open question, at the very least, this direction could lead to insights on the functionality of large language models by understanding how NTP maps language sparsity patterns to word/context representations. Importantly, such characterizations could also investigate linguistic regularities such as word analogies (e.g., representations where king-man+woman=queen). This relates to generalization, since the ability of a model to produce representations for which such arithmetic operations hold has been linked experimentally to better downstream generalization [Mik+13].
>
> [SK22] Stability vs Implicit Bias of Gradient Methods on Separable Data and Beyond
>
> [Mur+23] Grokking of Hierarchical Structure in Vanilla Transformers
>
> [Moh+23] Grokking modular arithmetic can be explained by margin maximization
>
> [Mor+24] Feature emergence via margin maximization: case studies in algebraic tasks
>
> [Mik+13] Efficient Estimation of Word Representations in Vector Space
>
> **Q: text too dense.**
>
> Well received. It makes sense to slightly shorten Sections 6 and 7 by moving some parts to the appendix.
>
> **Q: lines 32-34.**
>
> We appreciate the careful read. We will correct the typo by replacing "when" with "of" in Line 32.

---

> > ### Comment · Reviewer_kgdu · 2024-08-13
> >
> > I apologise for my delayed response, which was due to force majeure.
> >
> > > We conjecture that the source of inductive bias for this structural grokking phenomenon is actually the NTP training, and that the framing of NTP in our paper might provide the right tools to study this since: [...]
> >
> > In case my input is useful, I agree with this conjecture.
> >
> > I would like to sincerely thank you for your comment. It taught me many things. In general, I believe that the paper is a solid contribution and should be accepted.

---

> > > ### Author Response · Authors · 2024-08-13
> > >
> > > Thank you for your kind words and for sharing your thoughts on the conjecture—this is encouraging!
> > > We appreciate your time and support.

---

### Official Review · Reviewer_6D5d · 2024-07-14

**Soundness:** 3
**Presentation:** 3
**Contribution:** 3
**Rating:** 7
**Confidence:** 2

**Summary:**

This paper studies the structural properties of the solutions selected by gradient-based optimizers among the many possible minimizers of the NTP objective, the central challenge being to discern the "implicit bias" of the optimizer towards particular solutions.

**Strengths:**

- The paper is generally well written, and the notation is very clear.

- The paper provides a a very interesting starting point for studying the solutions found by gradient descent in NTP settings

**Weaknesses:**

While the paper provides a a very interesting starting point for studying the solutions found by gradient descent in NTP settings, it's not very clear whether margin maximization practically corresponds to any meaningful takeaway in language modeling.

**Questions:**

Just the remark in the weaknesses.

**Limitations:**

Yes

---

> ### Author Rebuttal · Authors · 2024-08-06
>
> Thank you for your time and for the positive feedback and score.
>
> **Q: While the paper provides a a very interesting starting point for studying the solutions found by gradient descent in NTP settings, it's not very clear whether margin maximization practically corresponds to any meaningful takeaway in language modeling.**
>
> We agree that this paper is a starting point and indeed we are currently working on extending it further. Below we’d like to share some of our early-stage thinking to help convey our vision for this line of work and its relevance to language modeling.
>
> A first idea is to use the problem formulation and convergence framework as a basis for characterizing the geometry of word and context representations of language models trained with NTP. In our paper, we take the initial step by fixing the context representations  and characterizing word embeddings. An idea to also explicitly account for context representations is to assume that context embeddings are freely optimized together with word embeddings. We briefly discuss this in Lines 361-7. An advantage of this approach is that it circumvents the complexities of a specific architecture (e.g., transformers) and isolates NTP from it. However, this requires assuming a large enough model with sufficient expressive power to generate such unconstrained embeddings. Recall that our current framing of the NTP paradigm as a sparse soft-label classification over distinct contexts, together with identifying the necessary and sufficient conditions for reaching the entropy lower bound, continue to hold in this setting. Thus, the technical question that future work needs to address is extending the convergence analysis to the bilinear setting where both $W$ and $h_j, j\in[m]$ are optimized. Our conjecture, pending further analysis, is that the convergence result will be of similar flavor: when projected to an appropriate data subspace, $F$, the word and context embeddings converge in direction to the solution of an appropriately defined margin-maximization problem. But, unlike the linear case where $F$ depends both on the embeddings and on the sparsity patterns of the language data, in this new setting, it is natural to expect that $F$ would only depend on the latter.
>
> It follows then that the (corresponding) margin maximization program would establish a direct mapping between language data statistics as encoded in the sparsity patterns of the training data and the geometry of representations. We deem this interesting in its own right. Moreover, down the road, we hope this might also help enhance model interpretability and explainability, as well as provide a way to algorithmically (e.g., by modifying the CE loss) mitigate unfavorable imbalances in language data (e.g., rare words/contexts). Indeed, characterizing the geometry of word representations has a long history in NLP literature. This dates back to at least the work [LV14], which studies word geometry of the Skip-gram with negative sampling objective in word2vec. This has been used to provide insights on ‘word analogies’ and inspire algorithms that modify the geometry of representations (e.g., making them more isotropic [Aro+16]) towards improving linguistic regularities [MBV17]. More recently, there are many works of similar flavor on modern transformer models trained with NTP, but to the best of our knowledge, most of these are heuristic/experimental, often resulting in contradictory claims. We see an opportunity for a theoretical framework to complement such work.
>
> Additionally, we envision that the results can be used to gain insights into how language models generalize. To give a more concrete example, it could be possible to use the results to theoretically investigate the empirically observed phenomenon of ‘Grokking of Hierarchical structure’, i.e., the ability of models to infer hierarchical structures in language data when trained far beyond the point of saturating the training accuracy [Mur+23]. While [Mur+23] report this phenomenon in transformers, we conjecture that the source of this structural grokking phenomenon is actually the inductive bias of NTP training. Various reasons lead us to believe that the framing of NTP in our paper might provide tools to study this: (1) the model of NTP as sparse soft-label classification fits synthetic language datasets such as Dyck, for which the structural-grokking experiments are reported in [Mur+23]. (2) It is experimentally suggested that structural grokking occurs only after extended training and only after the NTP loss saturates (i.e., when NTP-separability and compatibility conditions kick in). (3) The principle of margin-maximization has been very recently connected to the grokking phenomenon, although only in one-hot settings of modular arithmetic [Moh+23,Mor+24].
>
> Overall, while more work is needed to materialize the paper’s results into direct language modeling insights (e.g., the relation of representation geometry to language data statistics and how it impacts linguistic regularities, as well as generalization phenomena like structural grokking), we hope the above discussion (which we are happy to elaborate upon in the paper) convinces that it is a worthwhile endeavor.
>
> [LV14] Neural Word Embedding as Implicit Matrix Factorization
>
> [Aro+16] A latent variable model approach to psi-based embeddings
>
> [MBV17] All-but-the-top: Simple and effective post processing for word representations
>
> [Mur+23] Grokking of Hierarchical Structure in Vanilla Transformers
>
> [Moh+23] Grokking modular arithmetic can be explained by margin maximization
>
> [Mor+24] Feature emergence via margin maximization: case studies in algebraic tasks

---

> ### Author Response · Authors · 2024-08-13
>
> Dear Reviewer,
>
> We appreciate your support of our submission. We hope that our response regarding the potential takeaways of our study for language modeling sparked some interest. If you have any questions about these ideas, we are happy to elaborate. In any case, we look forward to hearing your feedback before the discussion period ends.

---

> ### Comment · Reviewer_6D5d · 2024-08-13
>
> Thank you for your response and for shedding light on how future directions and practical implications. Although this paper does not lie squarely within my area of expertise, I think it is a good paper, and will be raising my score to an accept.

---

> > ### Author Response · Authors · 2024-08-13
> >
> > Thank you again for your feedback and support.
> > We appreciate your decision to raise the score.

---

### Official Review · Reviewer_YBdf · 2024-07-17

**Soundness:** 2
**Presentation:** 2
**Contribution:** 2
**Rating:** 5
**Confidence:** 4

**Summary:**

This study investigates the structural properties of solutions chosen by gradient-based optimizers for next-token prediction (NTP), framing NTP as cross-entropy minimization across various contexts with sparse conditional probability distributions over a finite vocabulary. It focuses on the optimization bias of gradient descent (GD), characterizing how GD selects parameters that equate the logits’ differences of supported tokens to their log-odds.

**Strengths:**

This study enables deriving the data-entropy lower bound in NTP for understanding the optimization and generalization properties of NTP models.

**Weaknesses:**

The study's focus on linear models  analyzing CE loss for NTP may limit its novelty and applicability, making its contributions to the field appear unclear compared to existing research.

**Questions:**

Q.1 Please clarify the differences and advantages of your study compared to the following existing research. What new insights does this study provide, and why are they important? Specifically, while these existing studies highlight the critical role of attention in NTP, your study omits this aspect. Could you explain why it is still valid to disregard attention in your analysis?

Mechanics of Next Token Prediction with Self-Attention
Yingcong Li, Yixiao Huang, M. Emrullah Ildiz, Ankit Singh Rawat, Samet Oymak

Max-Margin Token Selection in Attention Mechanism
Davoud Ataee Tarzanagh, Yingcong Li, Xuechen Zhang, Samet Oymak

Transformers as Support Vector Machines
Davoud Ataee Tarzanagh, Yingcong Li, Christos Thrampoulidis, Samet Oymak

Q.2 When considering next-token prediction (NTP) using sequence data, distinct contexts might differ by only a single character and are expected to be interrelated. Does the assumption of independence and identically distributed (i.i.d) data in Eq.(2) not pose a problem in this scenario?

**Limitations:**

The limitations of this study include its reliance on the simplicity of the analyzed model, unclear distinctions and advantages over existing research, and its omission of key aspects such as the properties of attention mechanisms.

---

> ### Author Rebuttal · Authors · 2024-08-04
>
> Thank you for your review.
>
> Below, we clarify the distinctions from the references you mention and explain why our problem setting differs from studying self-attention/transformers, focusing instead on the NTP paradigm. While we have detailed these discussions in the submission, we repeat them here for your convenience, respecting the reviewing load.
>
> We hope these clarifications lead to a re-evaluation of your score!
>
> **Q1: Comparison to works on implicit bias in transformers:**
>
> We have discussed these references in detail in the submission. For your convenience, we summarize them below.
>
> [44] Mechanics of Next Token Prediction with Self-Attention - Li et al.
>
> [79] Max-Margin Token Selection in Attention Mechanism - Tarzanagh et al.
>
> [80] Transformers as Support Vector Machines - Tarzanagh et al.
>
> In Lines 327-328, we explain that our convergence results relate to a conjecture in [79] about implicit optimization bias in transformers, though our findings are different in nature. This is further elaborated in Appendix B, Lines 648-667. The discussion there is self-contained and directly answers your questions. We reproduce it below:
>
> “*As already mentioned in Sec. 6, our work is closely related to  [79], where the authors investigate the implicit bias of self-attention in transformers. The insight put forth in the prequel [80] is that  softmax attention induces implicit-bias behaviors that bear similarities to vanilla implicit bias of one-hot prediction. Concretely, [79] studies GD optimization of one-layer self-attention with fixed decoder and one-hot binary classification. They show that, in the limit, GD finds attention weights that converge in direction to the solution of an SVM problem that separates optimal tokens
> from non-optimal ones. Their non-convex setting introduces locally optimal SVM directions to which GD may converge depending on initialization. Different to them, the NTP setting that we study involves predictions over multiple categories and is not one-hot. Also, while they fix the decoder, here, we fix the embeddings. In these respects their results are rather different. More similarities arise when [79] replace the linear decoder  with a MLP, which they note can induce multiple optimal tokens per sequence. This leads them to formulate a more general token-separating SVM program, which similar to ours confines the separation on a certain data subspace. However, the operational nature of the programs remains different as theirs optimizes attention weights and separates tokens within a sequence, while ours optimizes decoder weights and separates context embeddings based on their respective support sets. More importantly, while [79] only conjectures the convergence of GD to their general SVM program, we leverage convexity in our setting to prove an analogous statement rigorously. Eventually, as we move lower in our top-down approach and consider architecture-specific embeddings  generated by attention, we anticipate to see integration of our ideas with  theirs.*”
>
> Additionally, Lines 677-684 compare our work to Li et al. [44]:
>
> “*Upon completing this paper, we became aware of independent contemporaneous research by Li et al. [44] that also examines the implicit bias of self-attention with a fixed linear decoder in next-token prediction scenarios. Unlike our study which utilizes the widely adopted CE loss, their approach is based on log-loss, which renders the training loss convex, a similarity shared with our model despite the inclusion of self-attention. Both our results and those of Li et al. substantiate the conjecture posited by Tarzanagh et al. [79], albeit in very distinct settings. Notably, contrary to both [79] and [44], we unveil the optimization intricacies of the NTP  paradigm, even within the simplest linear settings.*”
>
> We believe these detailed comments clarify the distinctions to the above references. If you have further questions, please kindly let us know and we are happy to elaborate.
>
> **Q2: Could you explain why it is still valid to disregard attention in your analysis?**
>
> Our key message is that self-attention and next token prediction (NTP) are distinct. NTP, which involves predicting the next token given preceding tokens using cross-entropy loss, is used across transformer-based models, state-space models, and LSTMs (see footnote 3). Thus, studying NTP separately from transformers/self-attention is valid.
>
> By isolating NTP, we highlight essential aspects of the problem often overlooked (e.g., [47,50]). For example, modeling NTP in language settings as soft-label classification over sparse probabilistic labels and deriving conditions for reaching the entropy lower bound. These foundational aspects remain valid regardless of the embedding architecture (see Lines 192-195).
>
> Our convergence results assume fixed embeddings and training only classifier weights (aka word embeddings). In Section 7, we suggest two avenues to extend this analysis. First, studying architecture-specific embeddings, including those generated by self-attention (Lines 352-357). Second, exploring architecture-agnostic embeddings via the unconstrained features model (Lines 361-367). This approach leads to a model where both context and word embeddings are freely optimized, helping to understand the geometry of context and work embeddings once the NTP loss reaches the lower bound.
>
> **Q.3 distinct contexts might differ by only a single character and are expected to be interrelated. Problem with independence and identically distributed (i.i.d) data assumption in Eq. (2)?**
>
> There is no i.i.d. assumption in Eq. (2). Rather, Eq. (2) is a reformulation of the empirical NTP loss expressed in terms of distinct contexts, with the summation over these contexts.
> If we have two contexts that differ by a single character, they are still considered distinct. Thus, Eq. (2) applies to them as it corresponds to two different distinct embeddings, say $h_{j_1}$, $h_{j_2}$ with $j_1,j_2\in[n]$.

---

> ### Author Response · Authors · 2024-08-13
>
> Dear Reviewer,
>
> Did our response address your questions? Specifically, did it resolve your concern about how our work positions itself in relation to the papers you mentioned?
>
> We look forward to your feedback before the discussion period ends.
>
> Thank you for your time.

---

> > ### Comment · Reviewer_YBdf · 2024-08-13
> >
> > Thank you for your response, I will raise my score.
> >
> > On Q2.
> > Let me confirm the motivation.
> > Does your response mean that next token prediction can be separated from a specific model to understand its properties?
> > If so, can such a separated theory provide a unified explanation of the phenomena of a specific model?
> > In that case, it it better to experimentally demonstrate  the findings obtained in this paper by using multiple different models.
> >
> > On Q3.
> > When inputting a context in a sliding window, the sequence that corresponds to the previous answer is partially included in the context. Isn't this not i.i.d even if it is distinct?

---

> > > ### Author Response · Authors · 2024-08-13
> > >
> > > Thank you for your response! And we appreciate your willingness to raise the score.
> > >
> > > **On Q2:** Yes, that’s right. The thesis we aim to communicate is that NTP as a training framework has intrinsic properties worth investigating, independent of the underlying architecture. This is not to say that architecture doesn’t matter, but significant insights can be gained by focusing on NTP itself. Our paper highlights this idea by examining a fixed embeddings setting, which allows us to isolate NTP properties. We provide experiments in the paper for this setting. More broadly, we envision using the proposed problem formulation and convergence framework to characterize the geometry of word and context representations in language models trained with NTP. Specifically, we aim to answer: how do the statistics of language data map to the geometry of representations during training?
> > >
> > > In our paper, we take an initial step by fixing the context representations and characterizing word embeddings. To also account for context representations, one idea is to assume that context embeddings are freely optimized along with word embeddings. We briefly discuss this in Lines 361-367. An advantage of this approach is that it circumvents the complexities of a specific architecture (e.g., transformers) and isolates NTP from it, provided the model is large enough with sufficient expressive power to generate such unconstrained embeddings. Our current framing of the NTP paradigm as a sparse soft-label classification over distinct contexts, combined with identifying the necessary and sufficient conditions for reaching the entropy lower bound, still holds in this setting. Thus, the technical question for future work is extending the convergence analysis to the bilinear setting where both $W$ and $h_j, j \in [m]$, are optimized. We conjecture that this will lead to an appropriate margin maximization program that establishes a direct mapping between language data statistics, as encoded in the sparsity patterns of the training data, and the geometry of representations. We find this inherently interesting, and such characterizations could also investigate linguistic regularities, such as word analogies (e.g., representations where king-man+woman=queen). Additionally, they might help enhance model interpretability and explainability, and provide a way to algorithmically (e.g., by modifying the CE loss) mitigate unfavorable imbalances in language data (e.g., rare words/contexts).
> > >
> > > **On Q3:** You are correct. The contexts are not i.i.d., but this does not create any problems in the current formulation. E.g., as mentioned: There is no i.i.d. assumption in Eq. (2) and such contexts are still viewed by the optimization as distinct contexts. What the implicit bias viewpoint of NTP suggests is that the relationship between learned context representations depends on the sparsity patterns of their next-tokens. That is, if two contexts (regardless of their degree of partial inclusion) are followed by a similar set of words (even if the probabilities of each word differ for the two contexts), their representations will tend to be more aligned, and vice versa.

---

> > > > ### Author Response · Authors · 2024-08-14
> > > >
> > > > Did this answer your question?
> > > > Also, we would appreciate if you could kindly update your score, as you mentioned in your previous response.
> > > >
> > > > Thank you for your time

---

> > > > > ### Comment · Reviewer_YBdf · 2024-08-14
> > > > >
> > > > > Thank you for your comments. I raised the score.

---

### Author Response · Authors · 2024-08-12

Dear Reviewers,

With all due respect to your reviewing load, we would appreciate your feedback on our responses.

We hope that we have adequately addressed your questions and concerns, and we would be delighted to see this reflected in your evaluations.

However, if you have any remaining questions, we are more than happy to respond while the discussion period is still ongoing.

Thank you again for your time.

---

### Decision · Program_Chairs · 2024-09-25

**Decision:**

Accept (poster)

**Comment:**

The authors investigate the implicit structural bias introduced by the gradient descent optimizer with the cross-entropy objective in the context of next token prediction (NTP), a common approach used in large language models (LLMs). This work extends the analysis of implicit bias from one-hot classification to the NTP setting, where the target output for a given context is a (sparse) distribution over the vocabulary rather than a single one-hot vector. This adjustment reflects the multiplicity of potentially correct tokens given the same context in NTP tasks. The paper's analysis is restricted to the linear layer in NTP while assuming a fixed context embedding. The results appear to follow a similar line to prior work on implicit bias in soft-max attention, though, the authors highlighted some notable conceptual as well as mechanistic distinctions. The impact of the work can also be further improved by extending their analysis to include the attention mechanism, which would offer a more complete perspective on implicit biases within NTP models.

Overall, the paper offers useful insights into implicit bias in NTP settings. The impact can be further enhanced by broadening the analysis to account for the attention mechanisms.